# Liquid biopsy-based single-cell metabolic phenotyping of lung cancer patients for informative diagnostics

Ziming Li[1,11], Zhuo Wang[2,11], Yin Tang[3,4,11], Xiang Lu[5,11], Jie Chen[3], Yu Dong[3], Baojun Wu[3], Chunying Wang[3], Liu Yang[6], Zhili Guo[7], Min Xue[7], Shun Lu[1], Wei Wei [4,5,8] & Qihui Shi[2,9,10]

Accurate prediction of chemo- or targeted therapy responses for patients with similar driver oncogenes through a simple and least-invasive assay represents an unmet need in the clinical diagnosis of non-small cell lung cancer. Using a single-cell on-chip metabolic cytometry and fluorescent metabolic probes, we show metabolic phenotyping on the rare disseminated tumor cells in pleural effusions across a panel of 32 lung adenocarcinoma patients. Our results reveal extensive metabolic heterogeneity of tumor cells that differentially engage in glycolysis and mitochondrial oxidation. The cell number ratio of the two metabolic phenotypes is found to be predictive for patient therapy response, physiological performance, and survival. Transcriptome analysis reveals that the glycolytic phenotype is associated with mesenchymal-like cell state with elevated expression of the resistant-leading receptor tyrosine kinase AXL and immune checkpoint ligands. Drug targeting AXL induces a significant cell killing in the glycolytic cells without affecting the cells with active mitochondrial oxidation.

[1] Shanghai Lung Cancer Center, Shanghai Chest Hospital, Shanghai Jiao Tong University, 200030 Shanghai, China. [2] Key Laboratory of Medical Epigenetics and Metabolism, Institutes of Biomedical Sciences, Fudan University, 200032 Shanghai, China. [3] Key Laboratory of Systems Biomedicine (Ministry of Education), Shanghai Center for Systems Biomedicine, Shanghai Jiao Tong University, 200240 Shanghai, China. [4] Institute for Systems Biology, Seattle, WA 98109, USA. [5] Department of Molecular and Medical Pharmacology, David Geffen School of Medicine, University of California, Los Angeles, CA 90095, USA. [6] Shanghai Bone Tumor Institute, Shanghai General Hospital, Shanghai Jiao Tong University, 200008 Shanghai, China. [7] Department of Chemistry, University of California, Riverside, CA 92521, USA. [8] Jonnson Comprehensive Cancer Center, David Geffen School of Medicine, University of California, Los Angeles, CA 90095, USA. [9] Minhang Branch, Zhongshan Hospital, Fudan University, 201199 Shanghai, China. [10] Institute of Fudan-Minhang Academic Health System, Minhang Hospital, Fudan University, 201199 Shanghai, China. [11]These authors contributed equally: Ziming Li, Zhuo Wang, Yin Tang, Xiang Lu. Correspondence and requests for materials should be addressed to S.L. (email: shunlu@sjtu.edu.cn) or to W.W. (email: wwei@systemsbiology.org) or to Q.S. (email: qihuishi@fudan.edu.cn)

The current clinical treatment decisions in non-small-cell lung cancer (NSCLC) are primarily driven by tumor genetics. However, patients with similar driver oncogene mutations may have variable responses to the same treatment[1]. For example, the clinical decision making of EGFR-TKI is normally based upon tumor genotyping to identify the existence of EGFR sensitive mutations. But at least 20–30% of NSCLC patients with EGFR sensitive mutations do not respond or develop resistance rapidly to EGFR-TKI treatment[2,3]. The focus on genetic alterations may not fully explain the fact that some NSCLC patients have diverse responses to EGFR-TKIs even if they bear the same EGFR sensitive driver oncogenes and do not concurrently have other resistance-leading mutations[4]. Likewise, cytotoxic chemotherapy is the primary treatment strategy for NSCLC patients without driver oncogene mutations[3], but the response profiles to chemotherapy also vary across patients[3]. There is no simple and cost-effective method in the clinic that can predict therapy response prior to the onset of therapy or identify potential drug resistance when the patients are still benefiting from the therapy. The lack of effective approach for pre-identifying the non-responders and short-term beneficiaries poses a significant challenge in clinical decision making for NSCLC patients.

Change in metabolic activity is often a fast and reliable readout of tumor cells in response to a stressful condition, such as drug treatment. A successful drug engagement is normally accompanied by the reduction of the aberrant glycolytic activity of tumor cells with a potential metabolic program switch to mitochondrial oxidation[5,6]. Such rapid inhibition on glycolysis, assessed by [18F]fluorodeoxyglucose (FDG) uptake through positron emission tomography (PET), has been utilized as an in vivo predictive biomarker of drug response for brain cancer[7]. Increasing evidence reveals that tumor cells can uncouple glycolysis from the mitochondrial oxidation, allowing the use of additional fuel sources, such as amino acids and fatty acids, to meet their heightened metabolic needs[8–10]. The diverse metabolic dependencies have been observed in different patient tumors, between the primary and metastatic lesions of the same patient, as well as within distinct regions of the same tumor[11–15]. They have major implications for therapies targeting tumor metabolic vulnerabilities. However, few studies have investigated the clinical applications of the substantial metabolic diversity in tumors, including drug selection as well as prediction of therapy efficacy and resistance. Recent studies suggest that the diverse responses to targeted therapies across patients with the same driver oncogenes may be attributed to the adaptive reprogramming of cancer cells beyond genetic level, where cellular phenotypic and metabolic diversity that allows tumor cells to flexibly adapt to various stressful conditions during tumor progression may play an important role[16,17]. These results prompt us to interrogate whether diverse metabolic profiles of tumor cells across lung cancer patients may be related to their heterogeneous therapy responses.

Pleural effusion containing rare disseminated metastatic tumor cells represents a valuable surrogate for the tumor tissue biopsy and allows us to interrogate the metabolic state of patient tumor cells. Pleural effusion is a common complication and often the first sign of lung cancer patients[18,19]. Compared to pleural biopsy or thoracoscopic surgery, pleural thoracentesis is the least invasive approach for clinical diagnosis of pleural effusion after patients receive a positive computed tomography (CT) scan of lung lesions[18,20,21]. Although a substantial amount of lung cancer patients develop pleural effusion during their disease course, the clinical utilities of the effusion fluid are largely limited to cytopathological and cell block analyses for confirmation of malignant pleural involvement and metastasis[20]. The rare disseminated tumor cells (DTCs) in body cavity fluids and peripheral blood

contain rich biomolecular information, among which the phenotypic and functional characteristics of these cells may be utilized to assess or predict patient therapy responses[22–24]. However, metabolic phenotyping of rare DTCs in circulation or other body fluids has barely been explored in clinical biospecimens due to the lack of single-cell metabolic assay that can robustly identify and analyze these rare cells.

To this end, we develop and employ an on-chip metabolic cytometry (OMC) platform and fluorescent metabolic probes to perform metabolic phenotyping on the rare DTCs in pleural effusions across a cohort of 32 lung adenocarcinoma (LADC) patients that covers prevalent driver oncogenes and molecular subtypes[25]. We quantify the glucose uptake and mitochondrial oxidation activity of cells at the single-cell resolution, followed by single-cell sequencing to dissect the molecular signatures associated with distinct metabolic phenotypes of tumor cells in pleural effusions. This pleural effusion-based metabolic assay thereby allows us to establish informative connections to patient therapy responses and clinical performances, and to accurately predict potential non-responders and short-term beneficiaries prior to the onset of therapy. The molecular information extracted from these disseminate cells further permits us to identify clinically actionable strategy for patients with poor clinical outcomes.

## Results

**Single-cell on-chip metabolic cytometry.** As demonstrated in our previous report[22], we developed an OMC assay for high-throughput screening of rare metabolically active tumor cells in liquid biopsy samples through exploiting the elevated glucose metabolism of the malignant cells compared to benign cells. It showed superior performance in malignant pleural effusion (MPE) diagnosis over standard clinical approaches, and allowed accurate diagnosis for patient samples with inconclusive diagnosis using traditional cytopathological analysis. In this work, we further expanded the capability of the OMC assay for assessing both cellular glycolytic activity and mitochondrial oxidation at the single-cell resolution. In brief, the OMC assay is performed on a PDMS microchip with a total of 110,800 addressable microwells located in 400 numbered blocks (Fig. 1a, b). The on-chip metabolic phenotyping employs a fluorescent glucose analog 2-NBDG and a mitochondrial redox indicator $C_{12}$-Resazurin (C12R) to probe cellular metabolic activity (Fig. 1a). Prior reports have shown that 2-NBDG enters a cell via glucose transporters and is phosphorylated at the C-6 position by hexokinases I-II. The phosphorylated fluorescent metabolite, 2-NBDG-6-phosphate, remains in the cell until decomposition into a non-fluorescent form[26–29]. 2-NBDG assay has been shown to be consistent with gold-standard FDG assay in quantifying in vitro glucose uptake of cells without dead cell interference (Supplementary Fig. 1)[22].

To fluorescently assess the cellular mitochondrial oxidation activity, we use non-fluorescent C12R that acts as an intermediate electron acceptor in the electron transport chain (ETC) and can be reduced to a fluorescent $C_{12}$-Resorufin primarily by mitochondrial NAD(P)H and FADH2 (Fig. 1c). Resazurin has been used as an oxidation-reduction indicator in a variety of assays for assessing the cell viability and mitochondrial metabolic activity[30–32]. Although resazurin reacts with various cell reducing components such as NADH, glutathione, amino acids in both non-enzymatic and enzymatic reactions, our results and other studies showed that it has much faster reaction kinetics with NAD(P)H compared to other major cellular reducing components, such as glutathione, in a short incubation time (Fig. 1d)[33,34]. For this reason, in live cells, the reduction of resazurin is primarily attributed to different oxidoreductase enzyme systems that use NAD(P)H as the primary electron

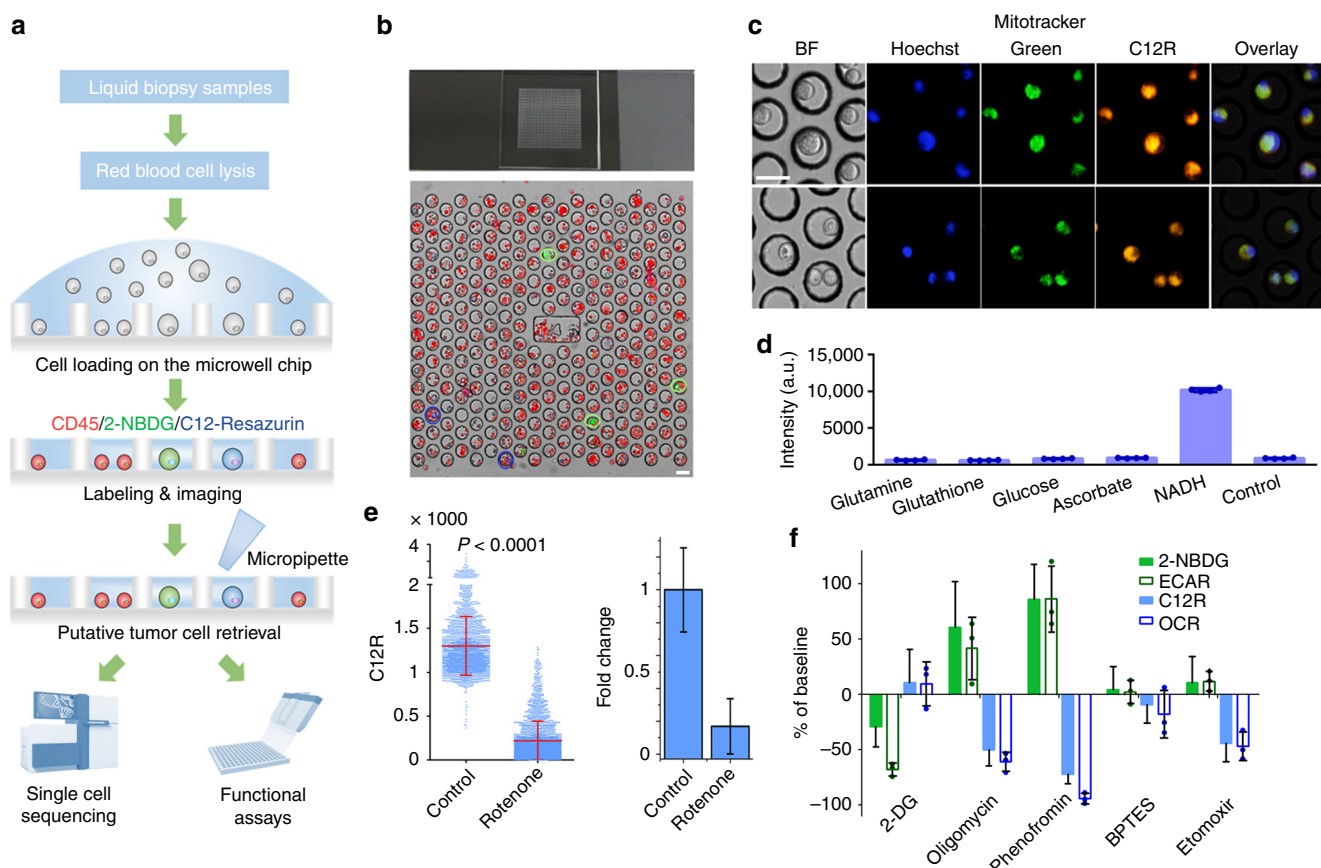

**Fig. 1** Platform and metabolic markers of the on-chip metabolic cytometry. **a** The working flow of the single-cell on-chip metabolic cytometry assay on pleural effusion samples based upon triple fluorescence staining (2-NBDG/C12R/CD45). Metabolically active tumor cells are retrieved individually for DNA and RNA sequencing, as well as other functional assays. **b** Top, a picture of the PDMS microwell chip; Bottom, the bright field and fluorescence composite image of a representative block in the microchip (scale bar, 30 μm). The fluorescence signals of CD45 (Cy5), 2-NBDG (FITC), and C12R (TRITC) are shown in red, green, and blue, respectively. Putative metabolically active tumor cells are circled in green (for 2-NBDG[high]) and blue (for C12R[high]). **c** Co-location of mitochondria (mitotracker green) and C12R in A549 cells after 15 min incubation (scale bar, 30 μm). **d** C12R signal assessed within a set of abundant cellular reducing agents using 60 min incubation time, showing the minimal contribution to the measured signal from other common cellular reducing agents. Quadruplets were used to determine the error bars ($n = 4$ independent experiments, mean ± SD). **e** On-chip metabolic cytometry of A549 cells treated with DMSO control and rotenone. The data are represented as scatter plots and bar columns (control $n = 2469$ cells, rotenone $n = 2501$ cells, mean ± SD). **f** Relative 2-NBDG ($n = 8433, 3498, 18148, 18987$, and $17248$ cells from left to right, respectively), ECAR ($n = 3$ independent samples), C12R ($n = 9576, 3673, 10298, 14625$, and $11428$ cells from left to right, respectively), and OCR ($n = 3$ independent samples) readout changes of A549 cells in response to a set of metabolic inhibitors with respect to DMSO control (mean ± SD). For ECAR and OCR measurements, three replicates were performed in each experiment and each replicate represented the average of ten cycles of ECAR and OCR measurements (Also see Supplementary Fig. 2d). Source data are provided as a Source Data file

donor[32,33,35]. To further ensure a majority of the resazurin signal coming from mitochondrial NAD(P)H, we used a hydrophobic carbon tail ($C_{12}$)-modified resazurin (C12R) to facilitate the fast diffusion to mitochondria. A co-staining assay on the lung cancer cell line A549 with Mitotracker Green and C12R demonstrated a clear co-localization of C12R signal and cellular mitochondria (Fig. 1c). As a further validation, we treated lung cancer cells with ETC complex I inhibitor rotenone[36] and complex V inhibitor oligomycin[37]. The treated cells showed a significantly reduced C12R signal compared to untreated cells, which is consistent with a repressed mitochondrial oxidation activity (Fig. 1e, and Supplementary Fig. 2a). Addition of fuels and precursor metabolites that can feed into the TCA cycle (glutamine, pyruvate, lactate, etc.) in the culture media also elevated number of C12R[high] cancer cells relative to baseline control (Supplementary Fig. 2b, c). Furthermore, a side-by-side comparison between the OMC assay and the commercially available Seahorse assay confirmed that 2-NBDG and C12R readouts are consistent with extracellular acidification rate (ECAR) and oxygen consumption

rate (OCR) readouts, respectively, when we perturbed A549 cells with a series of metabolic inhibitors to repress either glycolysis or mitochondrial respiration (Fig. 1f; Supplementary Fig. 2d). Taken together, these results indicate that 2-NBDG and C12R can be used as robust surrogates for assessing cellular glucose uptake and mitochondrial oxidation activity, respectively without mutual interference (Supplementary Fig. 3).

Figure 1a shows the workflow of the OMC assay from MPE sample processing to the rare cell retrieval. Red blood cells in MPE are firstly removed by osmotic cell lysis. Then around 500,000 nucleated cells are resuspended in Hank's balanced salt solution (HBSS) and incubated with allophycocyanin (APC)-labeled anti-CD45 antibody, 2-NBDG and C12R for 15 min. Cell suspension is applied onto a PDMS-based microwell chip followed by washing and imaging by a high-speed fluorescent microscope to capture a total of 728 images in ~10 min (Fig. 1b). A computational algorithm (MetaXpress software, Molecular Devices) analyzes the images at the single-cell resolution, identifies putative metabolically active tumor cells based on

calculated fluorescence cut-offs (Supplementary Fig. 4, see Methods for cut-off determination), and reports the corresponding microwell addresses. Target cells are retrieved individually by a micromanipulator based upon recorded addresses for single-cell sequencing of driver mutations, copy number variation (CNV), and transcriptome profiles (Supplementary Fig. 1c). Metabolically active cells found to harbor the same driver mutation as the primary lesion are confirmed to be malignant cells. For wild type (WT) primary lesion, the CNV profiles of those putative cells are evaluated to confirm the malignancy. The protocol only takes about 20 min from MPE sample processing to metabolic phenotyping via staining. Such a short duration helps retain the original metabolic profiles of those rare tumor cells with minimal perturbation.

**Heterogeneous metabolic reliance of DTCs in MPE samples.** To interrogate the metabolic phenotypes of the rare DTCs, we analyzed a cohort of MPE samples from 32 LADC patients through the OMC assay (Fig. 2). The 32 patients (except for patient-21) were diagnosed to bear stage IV LADC with different genetic background, driver mutations, and treatment history (Fig. 2a; Table 1; Supplementary Table S1). All of them developed MPE. We collected their MPE for metabolic phenotyping as well as downstream single-cell sequencing. The patients then received appropriate clinical treatments tailored to their own disease status (Supplementary Table S1). Since the reported average survival time for stage IV LADC patients with MPEs is around 6 months[38,39], We therefore chose 5–7 months after the MPE draw as a follow-up time point for review of patient performance, therapy response, and survival (Fig. 2a).

The scatter plot reports 2-NBDG and C12R fluorescence intensity of all CD45$^{neg}$ cells in the MPE sample from patient 1 (P1) (Fig. 2b). Measurement of CD45$^{pos}$ leukocytes was used to generate the cut-offs for identification of 2-NBDG$^{high}$ and C12R$^{high}$ cells (See Methods). Under these cut-offs, four subsets, including CD45$^{neg}$/2-NBDG$^{high}$/C12R$^{low}$ cells (2-NBDG$^{high}$ for short in the following text), CD45$^{neg}$/2-NBDG$^{low}$/C12R$^{high}$ cells (C12R$^{high}$ for short), CD45$^{neg}$/2-NBDG$^{high}$/C12R$^{high}$ cells (double positive), and CD45$^{neg}$/2-NBDG$^{low}$/C12R$^{low}$ cells (double negative), are thereby identified (Fig. 2c). The first three subsets are putative metabolically active tumor cells for subsequent genomic and transcriptomic analyses. The double negative subset may include normal epithelial cells, reactive mesothelial cells, as well as dying tumor cells with diminished metabolic activity. A total of 66 CD45$^{neg}$ metabolically active cells (per 500,000 input cells) were identified in P1, and 34 of them were isolated for single-cell sequencing. Among these 34 cells, 18 out of 20 (90%) 2-NBDG$^{high}$ cells, 7 out of 10 (70%) C12R$^{high}$ cells, and 3 out of 4 (75%) double positive cells were found to harbor an in-frame deletion (p. E746_A750del) in exon 19 of *EGFR*, consistent with the mutation status found in the primary lesion (Supplementary Fig. 5; Supplementary Table 2). The WT metabolically active cells exhibited similar CNV patterns as those *EGFR*$^{19Del}$-mutant cells, confirming their malignant involvement (Supplementary Fig. 6).

As surrogates of cellular glycolytic activity and mitochondrial oxidation, 2-NBDG and C12R intensities are consistently correlated in cancer cell lines with correlation coefficients >0.5 (Supplementary Fig. 7). However, such correlation disappears among the DTCs in patient MPE samples. The CD45$^{neg}$, metabolically active cells in the MPEs are either highly glycolytic (2-NBDG$^{high}$ cells) or having a high mitochondrial oxidation activity (C12R$^{high}$ cells) with few cells in double positive phenotype (Fig. 2d; Supplementary Figs. 8 and 9). This holds true for the entire cohort of 32 patients where no single patient

has a large percentage of double positive cells (Fig. 2d; Table 1). The two metabolic phenotypes showed selective sensitivities to inhibition of glycolysis and mitochondrial respiration, respectively (Fig. 2e). The majority of these metabolically active cells are confirmed to be malignant cells by sequencing (Supplementary Fig. 5). The uncoupling of glycolysis from the mitochondrial oxidation was further confirmed to exist in tumor cell populations isolated from primary tumor tissues (Supplementary Fig. 10) and may bear important implications.

**Prediction of therapy response and performance by the OMC.** While the MPE samples from patients with different driver oncogenes and treatment history had diverse metabolic phenotypes (Fig. 2d; Table 1), we found that the ratio of the number of 2-NBDG$^{high}$ cells to that of the C12R$^{high}$ cells, denoted as N/R ratio (2-NBDG/C$_{12}$-Resazurin), was predictive for the patient performance upon follow-up 5 to 7 months later after the MPE draw. Herein, P6, P21, and P32 were excluded in the subsequent analyses due to lack of appropriate treatment, lack of determined diagnosis, and lack of follow-up information, respectively (Table 1; Supplementary Table 1). The patient performance was evaluated by the ECOG score where 0 means asymptomatic and 5 means death. The higher the score, the worse the patient performance. The N/R ratios have a strong positive correlation ($r = 0.791$) with the ECOG scores upon follow-up (Fig. 3a), which indicates the prognostic capacity of this ratio.

ECOG score is a patient-centric score based upon physiological evaluation, which is in part the consequence of the patient's therapy response. We further verified if the N/R ratio is also predictive for the drug response profile of patient tumors assessed by Response Evaluation Criteria In Solid Tumors (RECIST 1.1). We employed a partial least square discriminant analysis (PLS-DA) to quantitatively assess the contributions of various factors to this tumor-centric RECIST criteria (Fig. 3b–d). PLS-DA is a multivariate linear regression model that seeks fundamental relations between the explanatory variable matrix (various clinical measurements) and the observation matrix (RECIST specification; Supplementary Table 3)[40]. Due to the relatively small sample size, we further categorized the partial response (PR) and stable disease (SD) as positive response (P), and progressive disease (PD) and death as negative response (N). In the explanatory variables, in addition to the N/R ratio, we also considered the numbers of 2-NBDG$^{high}$, C12R$^{high}$, and double positive cells per 500,000 input cells of MPE, as well as the volume concentration of the metabolically active cells (Supplementary Table 3). The classification functions of the model produced a confusion matrix with 86.21% accuracy in the leave-one-out cross validation (Fig. 3b; Supplementary Table 4) and an ROC curve with an area under the curve (AUC) of 0.952 (Fig. 3d), demonstrating a decent model quality and classification accuracy.

To quantitatively evaluate the contribution of various variables to the model, we calculated their Variable Importance for the Projection (VIP) that measures the importance of an explanatory variable for the prediction of the patient response profile (Fig. 3c). Consistent with what we found using patient-centric ECOG scores, the N/R ratio is most influential variable (VIP > 1). The number of 2-NBDG$^{high}$ cells is the second most influential variable and other variables are not predictive (Fig. 3c). Taken together, we can conclude that, while these patients are having different driver oncogenes, treatment history, and subsequent clinical managements after MPE draw, the N/R ratio of the rare DTCs in the MPE sample is a good predictive factor for the patient drug response profile and physiological performance.

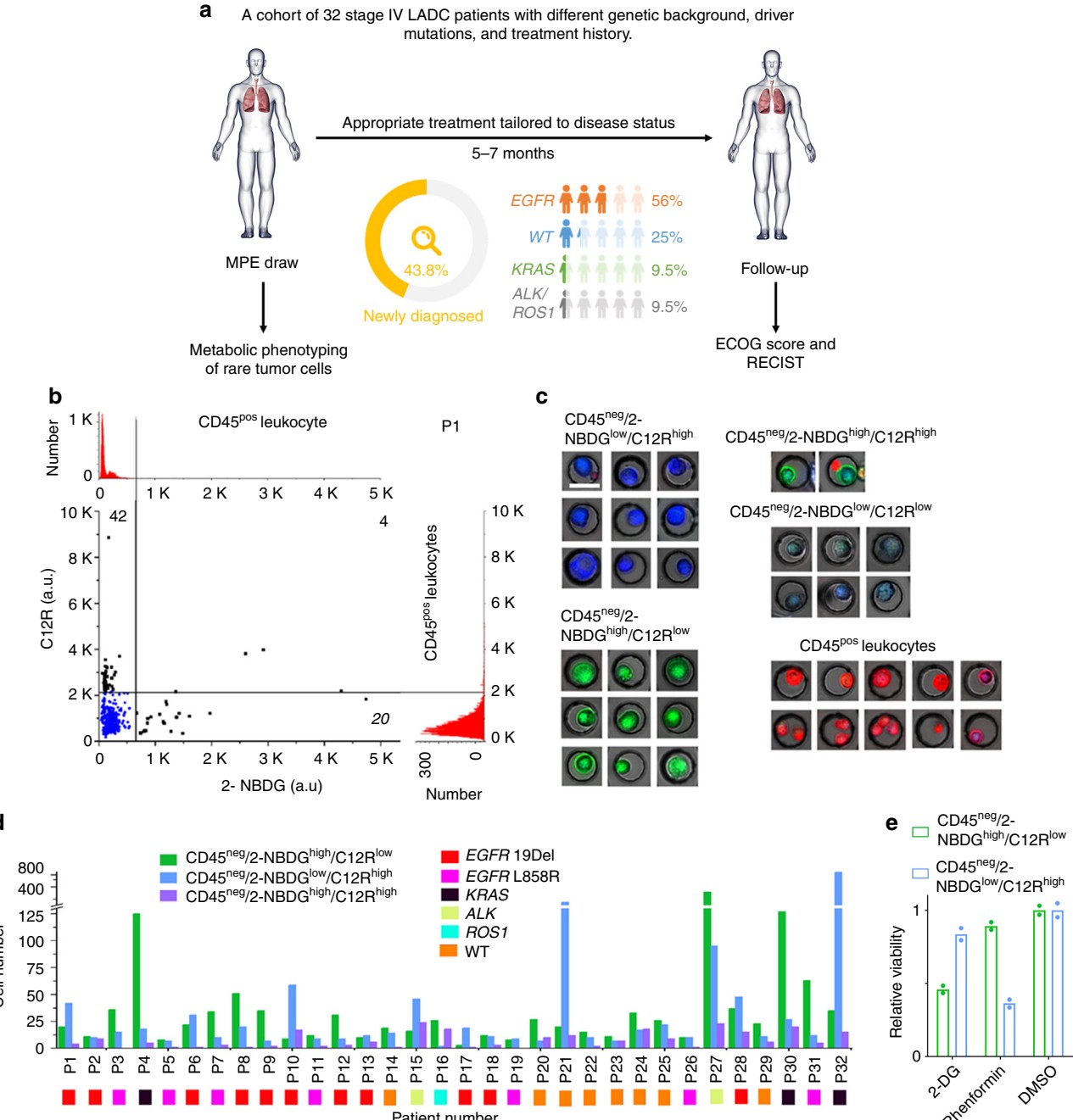

**Fig. 2** Metabolic phenotyping of rare disseminated tumor cells in MPE. **a** Illustration of study design and distribution of patient driver oncogene mutations. **b** Scatter plot generated from the OMC reports 2-NBDG and C12R fluorescence intensity of all CD45$^{neg}$ cells in MPE sample from P1. The histograms of 2-NBDG and C12R intensities of CD45$^{pos}$ leukocytes (red) in MPE are shown on the top and right to generate cut-offs for identification of 2-NBDG$^{high}$ and C12R$^{high}$ cells (black dots). 2-NBDG$^{high}$ and C12R$^{high}$ cells are gated out by five and three standard deviations above mean of CD45$^{pos}$ leukocytes, respectively. CD45$^{neg}$/2-NBDG$^{low}$/C12R$^{low}$ cells are displayed in blue dots. **c** Representative images of four subsets of CD45$^{neg}$ cells as well as CD45$^{pos}$ leukocytes. The images are overlaid by a bright-field image and three fluorescence images (CD45: red; 2-NBDG: green; C12R: blue, scale bar, 30 μm). **d** The number of CD45$^{neg}$, metabolically active cells that are categorized into three subsets across 32 LADC MPE samples. The oncogenic driver mutation associated with each sample is listed. **e** Relative viability of the two metabolically active cell populations in response to 2-DG and phenformin, respectively, with respect to the DMSO control ($n = 2$ independent samples, mean ± SD). (*$P < 0.05$; **$P < 0.005$). Source data are provided as a Source Data file

**The clinical utility of the OMC.** Although clinical trials over years have established many front-line experience and clinical factors (genetics, pathology, physiology, etc.) for anticipating LADC patient responses to standard-of-care treatments, there is no simple and cost-effective assay for predicting therapy responses and identifying non-responders and short-term beneficiaries prior to the onset of therapies. For example, in newly diagnosed patients with mutations that confer sensitivity to EGFR-TKIs, the established clinical factors may not effectively distinguish patients who bear *EGFR* sensitive mutations but do not respond to EGFR-TKIs. However, the metabolic phenotypes of their MPEs, represented by the N/R ratio, can accurately

**Table 1 Clinical metadata and on-chip metabolic cytometry data of LADC patients[a]**

| No. | Age range | Histology[b] | Stage | Driver mutations detected by OMC | Driver mutations in cell blocks[c] | Survival (months)[d] | PE volume (mL) | 2-NBDG$^{High}$/C12R$^{Low}$ | 2-NBDG$^{Low}$/C12R$^{High}$ | 2-NBDG$^{High}$/C12R$^{High}$ |
|---|---|---|---|---|---|---|---|---|---|---|
| 1 | 71–80 | ADC | IV | $EGFR^{19Del}$ | $EGFR^{19Del}$ | 20.6 (alive) | 22 | 20 | 42 | 4 |
| 2 | 51–60 | ADC | IV | $EGFR^{E19-A750P}$ + $EGFR^{19Del}$ | Wild type | 19.6 (alive) | 4.9 | 11 | 10 | 9 |
| 3 | 61–70 | ADC | IV | $EGFR^{L858R}$ | $EGFR^{L858R}$ | 15.6 | 47.6 | 36 | 15 | 0 |
| 4 | 71–80 | ADC | IV | $KRAS^{G12C}$ | $KRAS^{G12C}$ | 0.8 | 40 | 125 | 18 | 5 |
| 5 | 41–50 | ADC | IV | $EGFR^{L858R}$ | NA (Lung; $EGFR^{L858R}$) | 7.13 | 80 | 8 | 7 | 1 |
| 6 | 71–80 | ADC | IV | $EGFR^{19Del}$ | $EGFR^{19Del}$ | 9.1 | 3.9 | 22 | 31 | 1 |
| 7 | 41–50 | ADC | IV | $EGFR^{L858R}$ | NA (Lung; $EGFR^{L858R}$) | 10.1 | 11.7 | 34 | 10 | 3 |
| 8 | 81–90 | ADC | IV | $EGFR^{E19-A750P}$ + $EGFR^{19Del}$ | $EGFR^{19Del}$ | 7.03 | 3.9 | 51 | 20 | 1 |
| 9 | 81–90 | ADC | IV | $EGFR^{19Del}$ | $EGFR^{19Del}$ | 1.93 | 2.8 | 35 | 7 | 2 |
| 10 | 41–50 | ADC | IV | $EGFR^{19Del}$ | $EGFR^{19Del}$ | 15.2 (alive) | 10.8 | 9 | 59 | 17 |
| 11 | 61–70 | ADC | IV | $EGFR^{L858R}$ | $EGFR^{L858R}$ | 7.23 | 83.3 | 12 | 9 | 2 |
| 12 | 61–70 | ADC | IV | $EGFR^{19Del}$ + $EGFR^{T790M}$ | $EGFR^{19Del}$ + $EGFR^{T790M}$ | 4.03 | 75 | 31 | 9 | 3 |
| 13 | 41–50 | ADC | IV | $EGFR^{19Del}$ | $EGFR^{19Del}$ | 16.0 (alive) | 50 | 10 | 12 | 6 |
| 14 | 81–90 | ADC | IV | Wild type | Wild type | 7.13 | 16.7 | 19 | 14 | 1 |
| 15 | 31–40 | ADC | IV | ALK | NA (Lung; ALK) | 15.9 (alive) | 2 | 16 | 46 | 24 |
| 16 | 41–50 | ADC | IV | ROS1 | ROS1 | 2.43 | 11.2 | 26 | 2 | 18 |
| 17 | 61–70 | ADC | IV | $EGFR^{19Del}$ | $EGFR^{19Del}$ | 14.8 (alive) | 117 | 3 | 19 | 1 |
| 18 | 51–60 | ADC | IV | $EGFR^{19Del}$ | $EGFR^{19Del}$ | 20.0 (alive) | 8 | 12 | 11 | 3 |
| 19 | 51–60 | ADC | IV | $EGFR^{L858R}$ | NA (Lung; $EGFR^{L858R}$) | 17.7 (alive) | 50 | 8 | 9 | 0 |
| 20 | 71–80 | ADC | IV | Wild type | Wild type | 3.03 | 40 | 27 | 7 | 10 |
| 21 | 71–80 | Diagnosis undetermined | IV | Wild type | Wild type | 3.03 | 68 | 20 | 167 | 12 |
| 22 | 61–70 | ADC | IV | Wild type | Wild type | 15.4 (alive) | 16 | 15 | 10 | 2 |
| 23 | 61–70 | ADC | IV | Wild type | Wild type | 10.3 | 90 | 11 | 7 | 7 |
| 24 | 61–70 | ADC | IV | Wild type | Wild type | 5.93 | 2.2 | 33 | 17 | 18 |
| 25 | 41–50 | ADC | IV | Wildtype | Wild type | 14.3 (alive) | 0.5 | 26 | 22 | 9 |
| 26 | 51–60 | ADC | IV | $EGFR^{L858R}$ + $EGFR^{T790M}$ | $EGFR^{L858R}$ + $EGFR^{T790M}$ | 14.3 (alive) | 18.8 | 10 | 10 | 1 |
| 27 | 41–50 | ADC | IV | ALK | ALK | 5.5 | 12.5 | 299 | 95 | 23 |
| 28 | 41–50 | ADC | IV | $EGFR^{19Del}$ | $EGFR^{19Del}$ | 5.9 (alive) | 0.3 | 37 | 48 | 15 |
| 29 | 71–80 | ADC | IV | Wild type | Wild type | 5.9 (alive) | 5 | 23 | 11 | 6 |
| 30 | 71–80 | ADC | IV | $KRAS^{G12C}$ | $KRAS^{G12C}$ | 3.6 | 0.9 | 127 | 27 | 20 |
| 31 | 41–50 | ADC | IV | $EGFR^{L858R}$ | $EGFR^{L858R}$ | 1.3 | 3 | 63 | 12 | 5 |
| 32 | 51–60 | ADC | IV | $KRAS^{G12C}$ | $KRAS^{G12C}$ | NA[e] | 1.5 | 35 | 914 | 15 |

[a]In all 32 patients involved in this study, 19 (59.375%) of them are male. cytological analysis failed to identify malignant cells in PE samples of four patients (P5, P21, P22, and P23), but OMC detected candidate metabolically active tumor cells. P5, P22, and P23 were diagnosed based on the biopsies of the primary tumors. However, for P21, biopsy of the lung lesion was clinically high-risk and not performed because of the advanced age of the patient and the small size of primary lesion, leading to an inconclusive diagnosis. The last three columns of the table denote numbers of CD45$^{neg}$, metabolically active cells per 500,000 input cells from PE samples

[b]Key for histology: ADC, lung adenocarcinoma

[c]Mutations and detection methods of cell blocks from MPE samples: EGFR (ARMS), ALK (IHC), ROS1 (RT-PCR);

[d]The survival time was calculated from the time that pleural effusion was drawn and measured

[e]This patient was most recently enrolled in this study and treatment-naïve

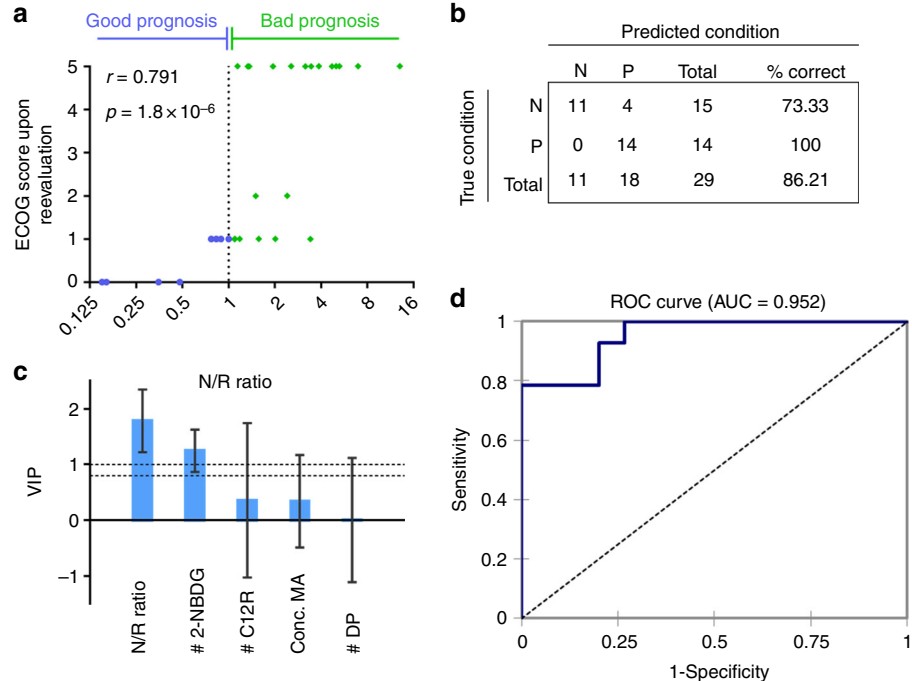

**Fig. 3** Partial least square – discriminate analysis (PLS-DA) for patient MPE samples. **a** The correlation between the N/R ratios and patient performance (ECOG scores upon follow-up) among 29 patient MPE samples. The Spearman correlation coefficient and the $p$ value are labeled. **b** Confusion matrix of the predicted patient response from the leave-one-out cross validation of the PLS-DA model. The overall accuracy of the model prediction is 86.21%. **c** Variable Importance of the Projection (VIP) of the explanatory variables. VIP values represent the predictive capacity of the various clinical measurements. These values are obtained from the single-component fit and error bars represent 95% confidence intervals. A border line is plotted to identify the VIPs that are greater than 0.8 for identifying the variables that are not (VIP < 0.8), moderately (0.8 < VIP < 1), or highly influential (VIP > 1) (# 2-NBDG: number of 2-NBDG$^{high}$ cells per 500,000 input cells; # of C12R: number of C12R$^{high}$ cells per 500,000 input cells; Conc. MA: concentration of metabolically active cells assessed by number of metabolically active cells in 10 mL MPE; # DP: number of double positive cells per 500,000 input cells). **d** The ROC curve based on the PLS-DA model prediction with an area under the curve (AUC) of 0.952. Source data are provided as a Source Data file

predict and segregate therapy response profiles for all our cases (Fig. 4a; Supplementary Fig. 11a), including newly diagnosed patients prior to the onset of the therapy.

In our patient cohort, we have 14 newly diagnosed patients from whom we analyzed the MPE samples prior to the onset of the 1st line therapy (purple dots in Fig. 4a and Supplementary Table 1). Among them, patients with predominantly glycolytic cells (N/R ≥ 2) in their effusions were all having a progressive disease and dead before the follow-up (diamond purple dots in Fig. 4a). In contrast, patients with predominantly mitochondrial oxidation cells (N/R ≤ 0.5) in their effusions were all partial responders with reduced tumor sizes upon follow-up. In addition, in the patients with a balanced metabolic phenotypes (0.5 < N/R < 2), four out of the five patients were having a stable disease upon follow-up (Fig. 4a). The segregation capability of the N/R ratio holds for patients with *EGFR* mutations (Fig. 4b; Supplementary Fig. 11b), other driver mutations, or WT tumors (Fig. 4a; Table 1; Supplementary Table S1). Even if the patients (P12 and P26) were diagnosed as having resistance-leading secondary mutation *EGFR$^{T790M}$* and receiving a third-generation EGFR TKI (Osimertinib) specifically targeting this mutation (black arrows in Fig. 4b), their N/R ratios could still segregate their drug response profiles.

In addition to predicting short-term therapy response profiles, the metabolic phenotyping has the potential to predict patient long-term survival as well. The patients with tumor cells of predominantly mitochondrial oxidation phenotype (N/R ≤ 0.5) or balanced phenotype (0.5 < N/R < 2) in their MPEs were having significantly longer survival time than patients with predominantly glycolytic cells (N/R ≥ 2) in the effusions (Fig. 4c). This holds true

regardless if we evaluated all the patients, or the newly diagnosed patients, or the patients with *EGFR* mutations (Fig. 4c). We further compared the metabolic phenotyping with patient $^{18}$FDG-PET scan for 3 newly diagnosed patients who performed PET/CT scan concurrently with the MPE collection. We found that the Maximum Standard Uptake Values (SUVmax) of their primary tumor mass were consistent with normalized 2-NBDG uptake values of the 2-NBDG$^{high}$ cells in their MPE samples (Fig. 4d, e). While the SUVmax has been reported to be negatively correlated with patient survival in NSCLC[41], compared to N/R ratios, the PET scan results are less predictive for patient therapy responses in our cases. For example, although P15 had the highest SUVmax in PET imaging and the highest relative 2-NBDG intensity of 2-NBDG$^{high}$ cell population among the three patients, this patient had more C12R$^{high}$ tumor cells present in the MPE and consequently a lower N/R ratio and better response and survival (Fig. 4e; Supplementary Table 1). In contrast, P30 with lowest SUVmax but highest N/R ratio had poorest response and shortest survival (Supplementary Table 1). Taken together, the metabolic phenotyping of the MPE samples could potentially serve as a simple assay for predicting the LADC patient response before the therapy start or identify potential therapy resistance when the patients are still benefiting from the therapy. It also holds the promise to score the current risk and long-term survival for lung cancer patients and potentially provides complementary information to $^{18}$FDG-PET imaging for more informative diagnostics.

**Molecular signatures of each metabolic phenotype.** To understand the molecular signatures that underlie the predictive

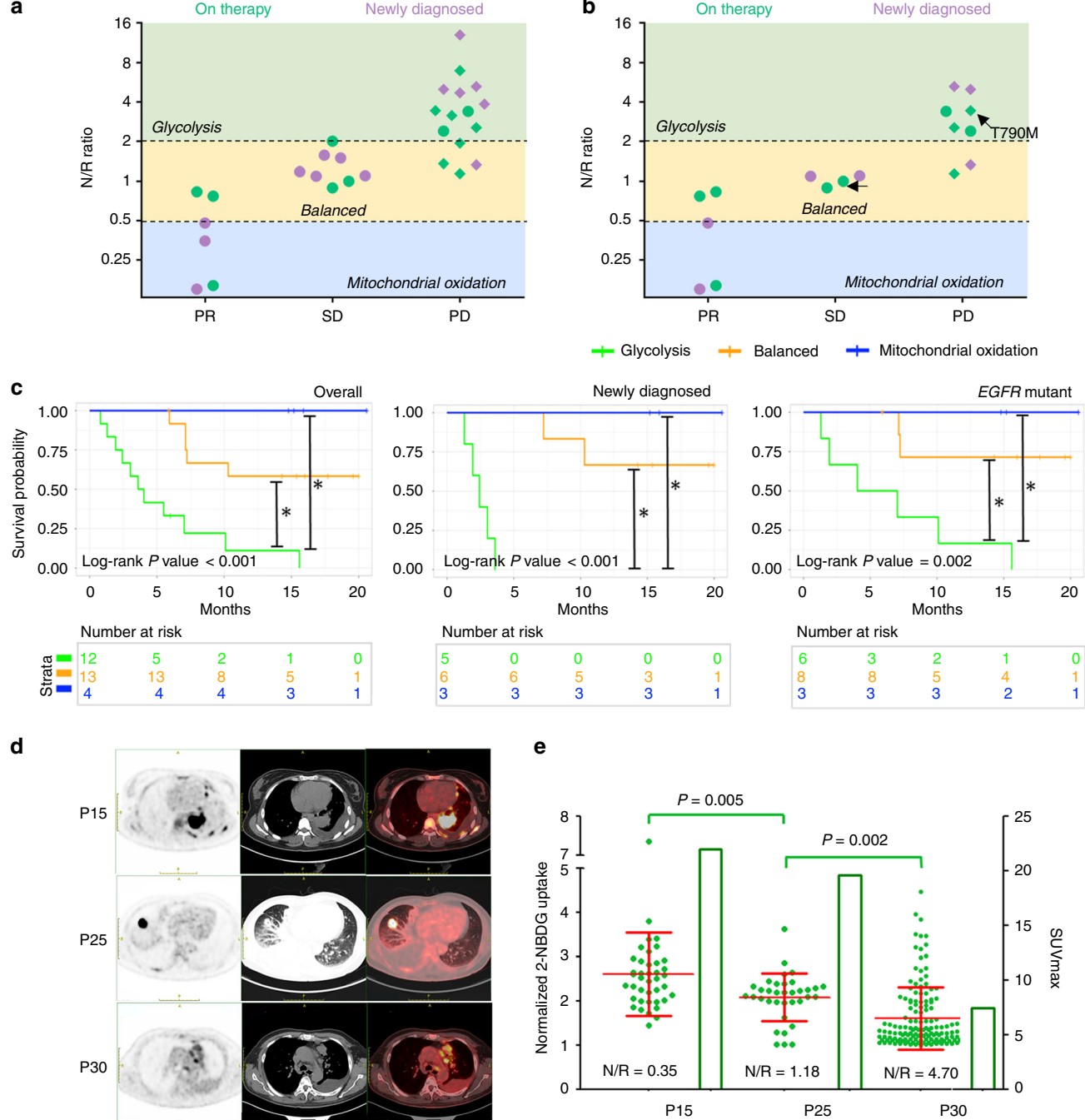

**Fig. 4** Therapy response and survival of patients predicted by the MPE metabolic phenotyping. **a**, **b** N/R ratios calculated by metabolic phenotyping of MPE samples can predict patient therapy response profiles evaluated by RECIST criteria upon follow-up for all the patients **a** and for *EGFR*-mutant patient **b**. The different metabolic phenotypes of MPE samples that are classified by the boundary lines at N/R = 0.5 and 2 can segregate patient response. The green dots denote patients who were on therapies at the MPE draw and metabolic phenotyping. The orchid dots denote newly diagnosed patients who were receiving first-line therapies after the MPE draw and metabolic phenotyping. The diamond dots represent the patients who were dead before the follow-up. PR: partial response, SD: stable disease, PD: progressive disease. **c** Kaplan–Meier survival curves for all the patients (left), newly diagnosed patients (middle), and EGFR-mutant patients (right) evaluated from effusion collection in groups of different metabolic phenotypes, respectively. Glycolysis phenotype, balanced phenotype, and mitochondrial oxidation phenotype are denoted as N/R ≥ 2, 0.5 < N/R < 2, and N/R ≤ 0.5, respectively. Vertical bars indicate patients censored at the cut-off date or loss of follow-up. The median survival time was significantly longer for patients in the mitochondrial oxidation group (not reached) and balanced group (not reached) than for glycolysis group (3.60 months, 95% CI: 1.93–7.03 months for overall, 2.43 months, 95% CI: 1.93–3.03 months for newly diagnosed, and 4.03 months, 95% CI: 1.93–10.10 months for *EGFR*-mutant) in all three cases. (Log-rank statistical test, *P < 0.05) **d** Maximum-intensity-projection PET images (left), axial CT (middle) and fused PET/CT images (right) showed variable FDG uptake in the pulmonary lesions across the three patients. **e** Normalized 2-NBDG uptake of 2-NBDG^high tumor cells in MPE samples overlaid with the SUVmax values from the PET images of the three patients. The 2-NBDG signals have been normalized to their respective cut-offs determined by 2-NBDG uptake of leukocytes for each patient (n = 40, 35, and 145 for P15, P25, and P30 respectively, mean ± SD). Source data are provided as a Source Data file

capability of the N/R ratio, we performed single-cell sequencing to profile the driver oncogene, CNV, and transcriptome of those metabolically active tumor cells from patient MPE samples. We first sought to identify if different driver oncogene mutations are associated with different metabolic phenotypes, and thus subsequently predictive for patients' drug response and performance. Unfortunately, we did not find any clear association between the driver oncogenes and cellular metabolic phenotypes in the 32 patient MPE samples (Table 1). For example, P9, P10, and P18 had the same driver oncogene $EGFR^{19Del}$ and received no prior treatment. However, they had diverse N/R ratios with P9 = 5, P10 = 0.15, and P18 ~1 respectively, which indicated that they had different dominant metabolic phenotypes in their metabolically active tumor cells from MPE samples. In addition, both P4 and P32 were bearing $KRAS^{G12D}$ mutations but with very different N/R ratios (P4 = 6.94; P32 = 0.04). The diverse metabolic phenotypes were also observed for patients with $ALK$ fusion alteration (Table 1). Therefore, no driver oncogene with metabolic phenotype preference was identified in our dataset. We next sought if additional protein-altering mutations, independent of known driver oncogenes, may contribute to the diverse metabolic phenotypes. To this end, we performed RNA-seq on the metabolically active tumor cells in 5 randomly selected patient MPE samples (Supplementary Data 1). For each patient, around 5–15 metabolically active tumor cells associated with each metabolic phenotype were retrieved and pooled together for sequencing. We called the mutations using the RNA-seq data and did not identify any protein-altering mutation incurred in a specific metabolic phenotype across all five patients (Supplementary Data 2). A small number of nonsynonymous mutations were found in certain metabolic phenotype across 4 out of 5 patients. However, none of these mutations have been reported to be related to cellular glycolysis and mitochondrial oxidation (Supplementary Data 2).

We further interrogated if the cellular CNV profile is related to the metabolic phenotypes. To minimize the potential interference from different genetic background, we analyzed the CNV profiles of the metabolically active tumor cells from the MPE samples of three patients (P1, P6, and P10) with the same $EGFR^{19Del}$ driver oncogene mutation (Supplementary Fig. 8 and Supplementary Table 2). Although the CNV profile varied from patient to patient, both metabolic phenotypes showed relatively consistent CNV profiles in a specific patient without any identifiable pattern that could robustly segregate one metabolic phenotype from the other across the three patients (Fig. 5a; Supplementary Data 3). Taken together, both the mutational alterations and CNV profiles appear insufficient to explain the observed uncoupling of metabolic phenotypes in patient MPE samples.

The null results from genetic analysis prompted us to further inspect the transcriptomic profile associated with each metabolic phenotype. The transcriptome data of the five patients displayed substantial patient-to-patient heterogeneity. We queried the differentially expressed genes (DEGs) between the two metabolic phenotypes for each patient. The majority of the DEGs were patient-specific (Fig. 5b). A closer inspection of gene expression levels confirmed the elevated expression of glycolysis-related genes in 2-NBDG^high cells, and up-regulation of genes that encode core subunits of mitochondrial ETC complexes and genes involved in fatty acid β-oxidation in C12R^high cells across patients (Fig. 5c). However, no clear pattern in glutaminolysis-related genes was observed for segregating the two metabolic phenotypes (Supplementary Fig. 12). Consistently, 2-NBDG^high and C12R^high cells showed selective sensitivities to inhibition of glycolysis, mitochondrial respiration, and fatty acid oxidation, but not to inhibition of glutaminase (Supplementary Fig. 13).

Enrichment of the DEGs shared by at least by 4 out of 5 patients against several public databases by Enrichr revealed significant cellular functions and pathways that were differentially regulated in the two metabolic phenotypes (Fig. 5d)[42]. We listed the top two entries enriched by the DEGs up-regulated in each metabolic phenotype from the databases ranked by enrichment scores. Among them, proton transporting activities, E-Cadherin signaling, and integrin signaling[43,44] were enriched in the genes up-regulated in C12R^high cells, which suggested the elevated ETC activities and epithelial polarity in the mitochondrial oxidation phenotype (Fig. 5d; Supplementary Data 4 and 5). Meanwhile, genes up-regulated in 2-NBDG^high cells showed enrichment in ribosomal biogenesis, translation elongation, and mRNA processing. It has been recently reported that ribosome biogenesis contributes to epithelial-to-mesenchymal transition (EMT) and metastatic cancer progression[45]. The Gene Set Enrichment Analysis (GSEA) of the entire transcriptomic dataset supported that the genes involved in EMT, metastasis, and SOX9 targets were significantly enriched in 2-NBDG^high cells for a majority of patients (Fig. 5e; Supplementary Data 6). SOX9 has been reported to prompt cytoskeleton alteration, invasion, and EMT in several cancer types[46,47]. These enrichments therefore suggested a mesenchymal feature and elevated metastatic potential in the glycolytic phenotype. A further inspection of EMT-related genes revealed that cells in glycolytic phenotype were more mesenchymal-like with repressed expression of epithelial-related genes ($EPCAM$, $CDH1$, $KRT$, etc.) and elevated expression of mesenchymal-related genes ($CDH2$, $WNT5A$, $TGFBI$, etc.) (Fig. 5f)[48]. In contrast, cells in mitochondrial oxidation phenotype were having elevated epithelial-related genes and reduced mesenchymal-related genes (Fig. 5f). The glycolytic phenotype were also found to have elevated checkpoint ligands (PD-L1, PD-L2) compared to mitochondrial oxidation phenotype across patients (Fig. 5f; Supplementary Fig. 14)[49], which is consistent with the recent reports that PD-L1 regulates glucose utilization and prompts glycolysis in cancer cells[50,51] and echoing the previous observation of the association between EMT and PD-L1 expression in lung cancer[52,53].

**AXL as a drug target for patients with a large N/R ratio.** The overexpression of AXL – an EMT associated receptor tyrosine kinase – has been increasingly appreciated as a key drug resistance and tumor dissemination mechanism in a number of solid tumors[54–56]. We found a consistent up-regulation of $AXL$ gene expression in 2-NBDG^high cells (Fig. 5f), which prompted us to investigate if AXL inhibition could be beneficial for patients with high N/R ratios and poor prognosis. To this end, we first confirmed that 2-NBDG^high cells from MPE samples have elevated AXL protein expression compared with C12R^high cells and CD45^pos leukocytes (Fig. 6a; Supplementary Fig. 15), which was in line with the $AXL$ gene expression pattern of these two metabolic phenotypes (Fig. 5f). We then collected MPE samples from three patients with WT tumor in chemotherapy treatment (P29), untreated $EGFR^{19Del}$ tumor (P9), and $EGFR^{L858R}$ tumor that had developed resistance to EGFR targeted therapy (P3), respectively. All the three patients had dominant glycolytic 2-NBDG^high phenotype in their MPE samples with N/R ratios larger than 2 (Table 1). We segregated both the 2-NBDG^high and C12R^high cells from the MPE samples and treated them with an AXL inhibitor R428 for 12 hours. Consistent with our expectation, R428 treatment led to significant cell killing in the 2-NBDG^high phenotype for all three patients (Fig. 6b; Supplementary Fig. 16). In contrast, it had minimal cell killing effect in paired C12R^high cells from the three patient MPE samples

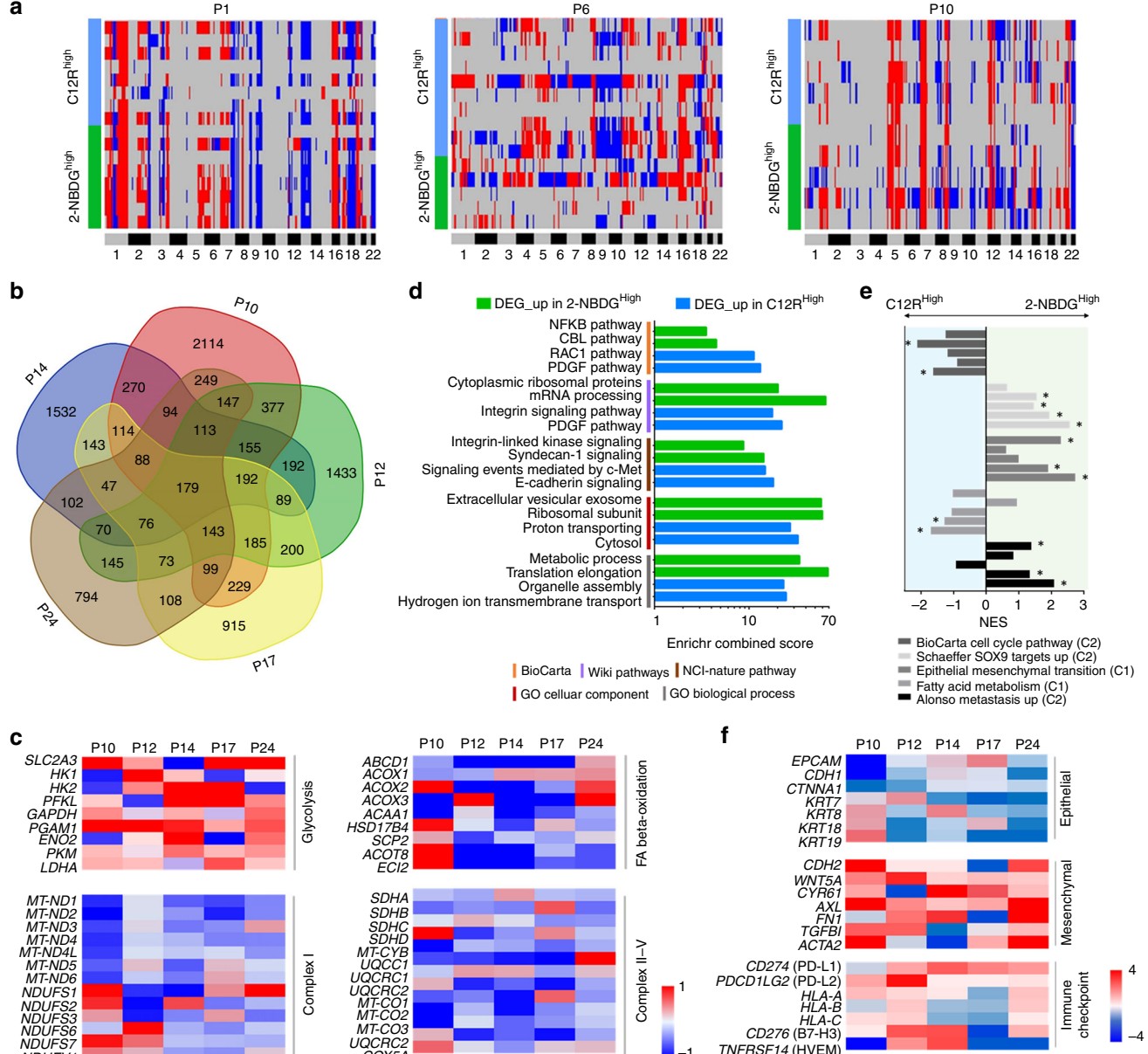

**Fig. 5** Molecular signatures associated with each metabolic phenotype. **a** Copy-number plot across the chromosomes for 2-NBDG[high] (green) and C12R[high] (blue) cells for three patient MPE samples, highlighting their genome-profile similarity between the two metabolic phenotypes. Blue indicates deletion and red amplification. **b** Venn Diagram of the differentially expressed genes (DEGs) between two metabolic phenotypes across five patient MPE samples. A total of 179 DEGs are shared among five patients. **c** Log2 fold change of gene expression levels between the two metabolic phenotypes (2-NBDG[high] vs C12R[high]) across five patients. Representative genes involved in glycolysis, mitochondrial ETC complexes, and fatty acid oxidation are listed. **d** Enrichment of the DEGs up-regulated in each metabolic phenotype and shared by at least 4 out of 5 patients against 5 representative public databases by Enrichr. The top two entries ranked by the combined scores from each database are plotted (see also Supplementary Data 5 for *p* values and enrichment scores). **e** Gene set enrichment analysis of 5 relevant pathways across the five patients (P10, P12, P14, P17, and P24 from bottom to top respectively). NES denotes normalized enrichment score (*$P < 0.05$ and FDR $q < 0.25$). **f** Log2 fold change of gene expression levels between the two metabolic phenotypes (2-NBDG[high] vs C12R[high]) across five patients. Representative epithelial and mesenchymal markers as well as immune checkpoint ligands and MHC Class I-associated HLA genes are listed

(Fig. 6c). These results suggest that an AXL targeted therapy may be considered for treating patients with high N/R ratio effusions.

## Discussion

Metabolic phenotyping of bulk tumor tissues has identified inter- and intra-tumoral regions with heterogeneous metabolic reliance[11,57,58], and has been utilized to reveal metabolic vulnerabilities and improve assessment of therapy response[59].

However, metabolic phenotyping of rare DTCs in body fluids with direct functional markers has not been achieved in part due to the lack of robust tools for identifying and analyzing these rare tumor cells in a complex biological fluid in a timely fashion. In this study, we demonstrate the clinical utility of such metabolic phenotyping through analyzing MPE samples of a group of LADC patients. Using the OMC assay, we identified 3 metabolically active phenotypes. 2-NBDG[high] cells represent a highly glycolytic phenotype that is ravenous for glucose with limited

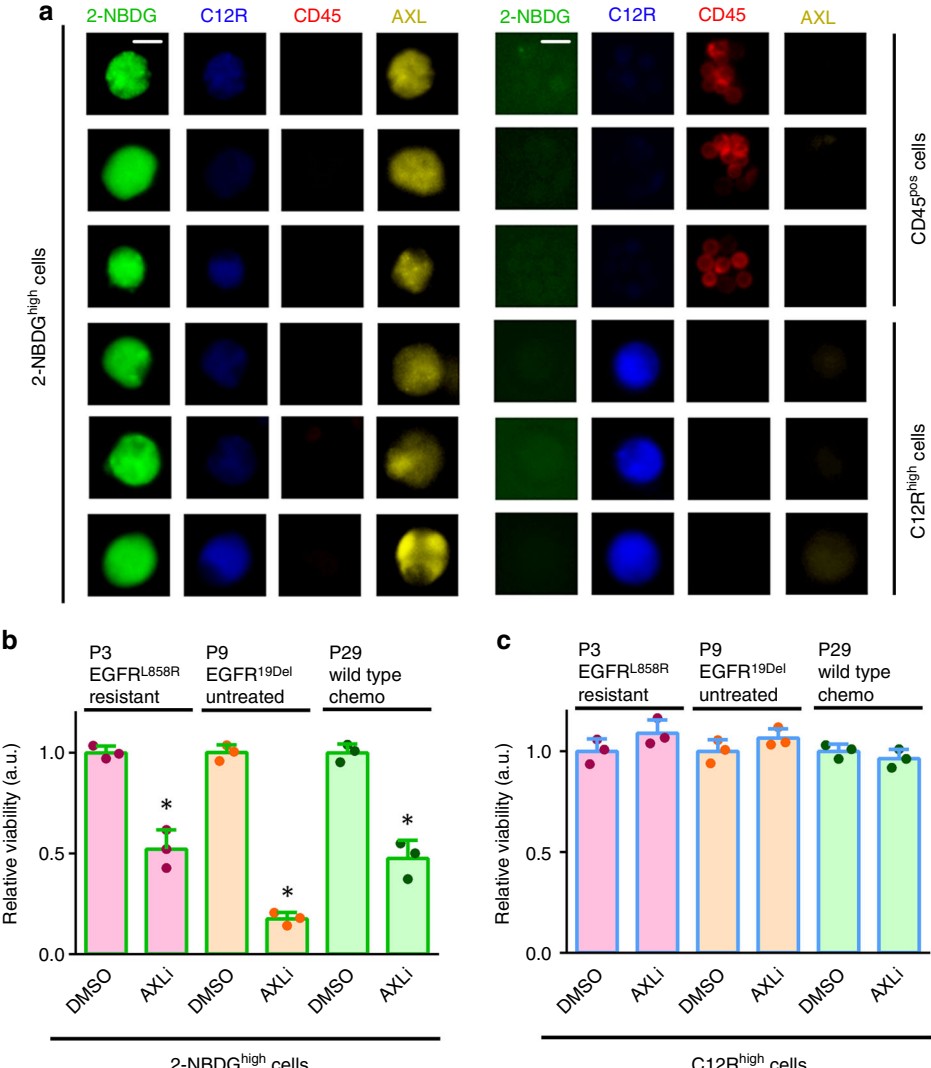

**Fig. 6** AXL as a potential drug target for the glycolytic phenotype. **a** Immunofluorescence staining shows detectable AXL receptor expression levels only on the 2-NBDG$^{high}$ cells in the MPE sample of P29 (scale bar, 10 μm). **b** Cell viability of 2-NBDG$^{high}$ cells upon AXL inhibition across three patient MPE samples with different genotypes ($n = 3$ independent experiments, mean ± SD, two-tailed Student's $t$-test, *$P < 0.05$). **c** Cell viability of C12R$^{high}$ cells upon AXL inhibition across three patient MPE samples with different genotypes ($n = 3$ independent experiments, mean ± SD). No statistically significant difference in cell viability is observed between drug and DMSO treated cells. Source data are provided as a Source Data file

mitochondrial oxidation. This is one of the most common features of cancer metabolism, as discovered by Otto Warburg in 1920s[60]. In contrast, C12R$^{high}$ cells have active mitochondrial oxidation but limited glucose uptake. They may either behave like normal cells by directing the majority of intake glucose to mitochondrial oxidation, or consume other fuels for bioenergetic and biosynthetic needs (Fig. 5c, e). The double positive phenotype covers cells with both enhanced glycolysis and mitochondrial oxidation. However, the percentage of these cells are relatively low in patient MPE samples (Fig. 2d). In contrast to cancer cell lines, we observed nearly mutual exclusive metabolic phenotype distribution in patient MPE samples (Fig. 2b, d; Supplementary Figs. 7 and 8). Such discrepancy may be attributed to the high functional heterogeneity of tumor cells growing in the pleural fluids that normally have more complicated microenvironment than the relatively homogeneous culture condition for cell lines. We showed that the cell number ratio of glycolytic phenotype to the mitochondrial oxidation phenotype can be used for assessing patient outcomes. Specifically, a patient with a high N/R ratio is likely to have a poor therapy response and physiological

performance upon follow-up 5–7 months later after MPE draw, as well as a shorter survival time (Figs. 3 and 4).

Our results did not identify a clear association between the N/R ratios and tumor mutations or CNVs, suggesting that the metabolic phenotypes may be regulated by epigenetic and/or transcriptomic programs beyond the genetic level. The predictive capability of this ratio was linked to the EMT programs, where we found that 2-NBDG$^{high}$ cells bore more mesenchymal-like features while C12R$^{high}$ cells were largely epithelial (Fig. 5c–f). Phenotypic composition and transition have been reported to be associated with the adaptive resistance to targeted therapies that allows tumor cells survive drug treatment prior to the establishment of genetically resistant clones[16,17]. Outstanding examples include the development of adaptive phenotypic response and drug tolerance of NSCLC[61], glioblastoma[17], melanoma[62,63], and circulating breast tumor cells[24,64] to chemotherapies and targeted inhibitions through EMT or similar phenotypic transition programs. We speculate that a very similar mechanism may contribute to our observation, where patients with MPE samples of predominantly glycolytic and mesenchymal cells may be more

tolerant to chemo- or targeted therapies, leading to poor prognosis[65]. While it is still unclear whether having mesenchymal-like cells in an MPE samples could reflect a mesenchymal feature of the primary lesion that is more difficult to treat, the similar mutual exclusive metabolic phenotypes were indeed observed in tumor cells isolated from primary tumor tissues (Supplementary Fig. 10). The excessive number of invasive mesenchymal-like cells in the pleural effusion might also contribute to the formation of distant metastases that cause poor clinical outcomes[66,67].

Nevertheless, for LADC patients who developed MPE, the metabolic phenotyping can provide information complementary to tumor genetics and other clinical factors. The N/R ratio appears to be a good predictive factor for patient clinical responses and survival. It holds the potential to identify non-responders and short-term beneficiaries of LADC patients prior to the onset of therapy (Fig. 4). We found higher AXL and checkpoint ligand expression levels (PD-L1, PD-L2, etc.) in 2-NDBG$^{high}$ cells (Figs. 5f and 6; Supplementary Figs. 13 and 14), which points to a possibility that patients with high N/R ratios may have higher AXL and checkpoint ligand expression in their tumor tissues. Given the observation that patients with high N/R ratios are less likely to get benefit from traditional chemo- or targeted therapies, alternative treatment strategies, such as inhibition of AXL in combination with the drugs that target the driver oncogenes or first-line checkpoint immunotherapy (anti-PD1, anti-PDL1, etc.), might be explored for this poor-prognosis population. Of course, these hypotheses need to be rigorously validated in a large patient cohort.

A limitation of this study is the relatively small cohort size and lack of patients treated by checkpoint blockade. While the patient population in this study covers major mutational subtypes of LADC, the number of KRAS-mutant or ALK fusion case is small. Although we did not see any exception in our dataset, the robustness and generality of the predictive capability of the N/R ratio in lung cancer requires further examination in a larger patient population before clinical translation, particularly for patients with low frequency driver oncogene mutations, patients receiving checkpoint inhibitors, and other tumor subtypes. The underlying molecular mechanism associated with each metabolic phenotypes also needs further validation in a more controllable tumor model system. Our results reported here motivate additional pre-clinical work to set a stage for such a large prospective clinical study.

## Methods
**Study design**. This pilot study tested the utility of metabolic phenotyping on rare DTCs in MPEs using single-cell OMC assays and fluorescent metabolic probes. Pleural effusion samples and remnant surgical tumor biopsies were collected from NSCLC patients in Shanghai Chest Hospital (Shanghai, China) with written informed consent and in accordance with guidelines and protocols that approved by the Ethics and Scientific Committees of Shanghai Chest Hospital. A majority part of the effusion sample from thoracentesis was sent to cytopathology for cytological examination and sequencing of cell blocks. The remnant sample usually consisted of 10–100 mL of cellular fluid and were processed immediately for on-chip metabolic assay. Metabolically active tumor cells with distinct metabolic signatures were retrieved individually by a micromanipulator for single-cell sequencing of driver mutations, CNVs, and transcriptome profiles. The cytopathological analysis on pleural effusions, sequencing of cell blocks, and clinical outcomes were performed independently and blinded to the operators. Thirty two patient samples were analyzed in this study. After MPE draw, the patients were then received appropriate clinical treatments tailored to their own disease status. Patient performance and therapy response were reevaluated after 5–7 months of MPE draw (Fig. 2a).

**Cell lines and reagents**. Human lung adenocarcinoma cell lines A549, NCI-H1650, NCI-H1975, and HCC827, human colorectal carcinoma cell line HCT116, human osteosarcoma cell lines MG63 and 143B were obtained from the cell bank of Chinese Academy of Sciences and routinely maintained in RPMI-1640 Medium (Life Technology, A10491-01) containing 10% FBS in humidified atmosphere of 5% CO2 and 95% air at 37 °C. Allophycocyanin (APC)-conjugated CD45 (clone HI30) and 2-

NBDG (2-(N-(7-Nitrobenz-2-oxa-1,3-diazol-4-yl)Amino)-2-Deoxyglucose) were purchased from Life Technologies. C12-Resazurin (IUPAC name of resazurin: 7-hydroxy-10-oxidophenoxazin-10-ium-3-one) obtained from Vybrant cell metabolic assay kit and FITC-conjugated goat-anti-rabbit secondary antibody were purchased from Thermo Fisher Scientific. Anti-PD-L1 (#BX00006, clone RR604) and its positive cell line were purchased from Biolynx. Hank's balanced salt solution (HBSS, no calcium, no magnesium, no phenol red), 0.25% Trypsin-EDTA, Dulbecco's modified eagle medium (DMEM, glucose free) and DMEM (glucose free, glutamine free, sodium pyruvate free) were purchased from Gibco. D-glucose, sodium lactate, sodium pyruvate, L-glutamine, DMSO (dimethyl sulfoxide) were obtained from Sigma-Aldrich. Oligomycin was purchased from Cell Signaling Technology. Rotenone, phenoformin, 2-Deoxy-D-glucose (2-DG), BPTES and etomoxir were purchased from SelleckChem. Cell strainers (70 μm, 100 μm) were purchased from BD Falcon. Poly(dimethylsiloxane) (PDMS) pre-polymer (Sylgard 184) was purchased from Dow Corning. Photoresist SU-8 2050 was purchased from MicroChem Corp. Single-cell whole-genome amplification kit was purchased from Qiagen. Taq DNA polymerase premix kit and TruePrep DNA Library Prep Kit V2 for Illumina were purchased from Vazyme Biotech. NEBNext dsDNA Fragmentase and NEBNext Ultra™ II DNA Library Prep Kit were purchased from New England Biolabs. ERCC RNA Spike-In Mix and Qubit dsDNA HS Assay Kit were purchased from Life Technologies. Agencourt AMPure XP was purchased from Beckman Coulter. SMART-Seq v4 Ultra Low RNA Kit was purchased from Clotech. All primers were synthesized by Life Technologies and listed in Supplementary Tables 2 and 5.

**Fabrication of microwell chip**. The microwell chip was fabricated in PDMS using standard microfabrication soft-lithographic techniques. A replicate for molding the PDMS was obtained by patterning a silicon wafer using photoresist SU-8 2050. The PDMS pre-polymer was mixed in a ratio of 10:1, and subsequently casted on this lithographically patterned replicate. After curing at 80 °C for 2 h, the PDMS component was separated from the replicate.

**Single-cell OMC assay of pleural effusion samples**. Typically, 10 mL of pleural effusion was filtered by a membrane with a pore size around 100 μm, followed by centrifuging at 500 × g for 5 min to separate cell pellets. 1 mL of red blood cell lysing buffer (BD) was then added to lyse red blood cells for 5 min. After centrifuging at 500 × g for 5 min, the nucleated cell pellet was resuspended in and washed with HBSS. After cell counting, ~500,000 cells were treated with 1 μl of APC-conjugated anti-CD45 antibody (BD Biosciences), 400 μM 2-NBDG and 1 μM C12R in glucose-free DMEM for 10 min in a cell incubator. Cell suspension was then applied onto a 3% Matrigel (BD Biosciences)-coated microwell chip as a monolayer and wait for 5 min in a cell incubator until cells sitting down in the microwells. All cells on the chip were extensively washed with cold PBS and DMEM. An ImageXpress Micro XLS Widefield High Content Screening System (Molecular Devices) scanned the chip and imaged all cells in bright field and three fluorescent colors (CD45: CY5, 2-NBDG: FITC, C12R: TRITC). MetaXpress software (Molecular Devices) analyzed the images and identified metabolically active tumor cells (CD45$^{neg}$/2-NBDG$^{high}$ or CD45$^{neg}$/C12R$^{high}$) based on the cut-offs generated from 2-NBDG and C12R fluorescence signals of CD45$^+$ leukocytes on the chip. The cut-off of 2-NBDG$^{high}$ cells is defined as mean plus five standard deviations of CD45$^+$ leukocytes, and the cut-off of C12R$^{high}$ cells is defined as mean plus three standard deviations of CD45$^{pos}$ leukocytes. The numbers of three metabolically active subsets (CD45$^{neg}$/2-NBDG$^{high}$/C12R$^{high}$, CD45$^{neg}$/2-NBDG$^{low}$/C12R$^{high}$, CD45$^{neg}$/2-NBDG$^{high}$/C12R$^{low}$) were recorded for each patient involved in this study. Some of metabolically active tumor cells in the microwells were individually retrieved using a XenoWorks Micromanipulator and trimethylchlorosilane (TMCS)-treated micropipettes, and then transferred into low binding PCR tubes (Axygen) for downstream single-cell sequencing.

**C12R reaction kinetics with major cellular reducing agents**. 100 μL of reducing agents (Glutamine 4 mM, Glutathione 16 mM, Glucose 12 mM, ascorbate 1.6 mM, NADH 1.2 mM in 10 mM PBS buffer) were mixed with 100 μL of diaphorase and 200 μL of C12Rz (2 μM in 10 mM PBS buffer), respectively and incubated at room temperature for 1 hour. The final concentration of the reducing agents are Glutamine 1 mM, Glutathione 4 mM, Glucose 3 mM, ascorbate 0.4 mM, NADH 0.3 mM. The fluorescence intensity was determined by a microplate reader (540 nm excitation and 590 nm emission). Then the experiment was carried out in quadruplets to determine the error range.

**Single-cell oncogenic driver mutation detection**. Single-cell whole-genome amplification (WGA) was performed on single cells using a REPLI-g Single Cell Kit (Qiagen). PCR for the target regions was performed using the primers listed in Supplementary Table 1 using 12.5 μl 2X Ex Taq DNA polymerase mix (Vazyme Biotech), 10 μM forward primer, 10 μM reverse primer, 0.2 μl WGA DNA. The PCR conditions used were: 95 °C for 3 min, followed by 30 cycles (95 °C for 30 s, 60 °C for 30 s and 72 °C for 30 s), a final extension at 72 °C for 5 min. The PCR products were analyzed with Sanger sequencing (Genewiz, Suzhou, China).

**Single-cell copy number variation detection**. To evaluate the WGA amplification coverage of single cells genome, we designed 22 pairs of primers (Genewiz, Suzhou,

China) to target 22 loci on different chromosomes (Supplementary Fig. 17). The primer sequences are listed in Supplementary Table 5. After WGA reaction of candidate tumor cells, QC PCR reactions were conducted at 95 °C for 3 min, followed by 30 cycles (95 °C for 20 s, 60 °C for 20 s and 72 °C for 30 s) and a final extension at 72 °C for 5 min. WGA products that passed QC were digested with NEBNext dsDNA Fragmentase (New England Biolabs) and 300–500 bp fragments were selected using Agencourt AMPure XP Beads (Beckman Coulter). We used 100 ng of DNA fragments as input to prepare sequencing libraries. Libraries were prepared by NEBNext Ultra™ II DNA Library Prep Kit for Illumina (New England Biolabs), fragments end-repair, 3′ adenylation and ligation according to the manufacturer's instructions. 0.8X Agencourt AMPure XP (Beckman Coulter) was used for purification and we then performed 9 cycles of PCR following the manufacturer's instructions, using PE5/7 primers (New England Biolabs). Agencourt AMPure XP (Beckman Coulter) was used for final library purification. The concentration of purified fragmented DNA or libraries concentration was measured with Qubit dsDNA HS Assay Kit (Invitrogen) in steps, and final libraries quantified by a 2100 Bioanalyzer (Agilent Technologies). Libraries were analyzed by Illumina HiSeq X Ten platform with 150 bp pair-end reads (Genewiz, Suzhou, China).

**Transcriptome sequencing of rare disseminated cells in MPE.** 5–15 metabolically active cells were retrieved and pooled together into low bind tube (Axygen) with 12.5 μl lysis buffer, which contained 10X Reaction buffer, ERCC (1:10$^6$, Invitrogen), 3′ SMART-Seq CDS Primer IIA, RNase Inhibitor plus DEPC-treated water. Single-cell transcriptome amplifications were performed using SMART-Seq v4 Ultra Low RNA Kit as described in the protocol for the kit (Clotech). The amplified cDNA products were purified with 0.8X Agencourt XP DNA beads (Beckman Coulter). The concentration of purified cDNA was quantified with Qubit dsDNA HS Assay Kit (Invitrogen), and libraries were then constructed with the TruePrep DNA Library Prep Kit V2 for Illumina (Vazyme Biotech) and quantified by a 2100 Bioanalyzer (Agilent Technologies). Libraries were analyzed by an Illumina HiSeq X Ten sequencer with 150 bp pair-end reads (Genewiz, Suzhou, China).

**Analysis of RNA sequencing data.** RNA sequences of tumor cells were aligned to the known human transcriptome (hg19) using HISAT2 (version 2.1.0)[68]. Reads that did not align or aligned to multiple locations in the genome were discarded. The hg19 GTF file from Ensemble was used to map. The reads count for each gene was the number of reads that were so mapped to that gene. This count was measured by HTSeq (version 0.8.0)[69], and Reads Per Kilobase Million (RPKM) value was used to quantify gene expression level. RPKM values from replicates of one metabolic phenotype of the same patient were merged and averaged.

For identifying tumor-specific protein-altering mutations, RNA-sequencing reads from one phenotype of the same patient were merged and aligned to the major chromosomes of human (hg19) using BWA (v0.7.16) with default options[70]. Then the duplicated reads were removed. We called SNPs using SAMTools (v1.4)[71] and BCFtools (v1.3)[72] with default options and filtered by SNPs called from matched leukocytes. An SNP was retained only when it was covered by least 5 reads in one of all samples.

Cuffdiff in Cufflinks package (version 2.2.1) was used to identify DEGs (false discovery rate < = 0.05)[73] between the two metabolic phenotypes. The DEGs up-regulated in each metabolic phenotype shared by at least 4 out of 5 patient MPE samples were enriched by Enrichr algorithm[42] against curated databases including BioCarta, WikiPathways, NCI-Nature Pathway, Gene Ontology (GO) cellular components and GO Biological process, respectively (Supplementary Data 5). The top 2 entries ranked by the Enrichr combined scores were listed in Fig. 5d.

Gene Set Enrichment Analysis was performed as described by as described by Subramanian et. al.[74]. In brief, genome-wide expression profiles from samples belonging to two labeled markers (2-NBEG or C12R) were used to perform this analysis. Genes were ranked based on the correlation between their expression and the class distinction by using metric of Signal2Noise. The enrichment results for MSigDB Hallmark and C2 gene sets were listed in Supplementary Data 6.

**Copy number determination and segmentation.** Sequence reads were aligned to the major chromosomes of human (hg19) using BWA (version 0.7.16)[75] with default options. To reduce whole-genome sequencing biases caused by difference of GC contents in the genome, the sequence depths of tumor cells were normalized by sequence depths from several normal white blood cells (WBC) (CD45$^+$)[76]. The CNV regions were identified as described previously[76,77]. Briefly, the likely diploid regions were determined using the hidden Markov model (HMM). The identified diploid regions were then used to provide a normalization factor for determining copy number. Similar copy numbers in adjacent chromosome regions were merged HMMcopy[78] (version 0.1.1) package.

**Metabolic activity assay.** A549 cells were cultured in the microwell chip and treated with 2 μM oligomycin (inhibitor of oxidative phyosphorylation,) for 6 h at 37 °C, followed by adding 2-NBDG to yield a final concentration of 600 μM. A microchip containing A549 cells with DMSO vehicle was used as the control. After 10 min assay of 2-NBDG, the chip was imaged with an ImageXpress Micro XLS Widefield High Content Screening System for measuring fluorescence intensity of 2-NBDG uptake. Rotenone (inhibitor of mitochondrial respiratory complexes I)

inhibition experiments were conducted in a similar way. A549 cells were on-chip treated with 5 μM rotenone for 6 h, followed by adding C12R to yield a final concentration of 1 μM. To investigate the influence of carbon sources on cell metabolism, A549 cells were treated with glucose (11 mM), glutamine (2 mM), sodium lactate (10 mM), and sodium pyravate (1 mM) for 2 h, respectively. The carbon sources (glucose, glutamine, sodium lactate, sodium pyravate) were dissolved in a DMEM medium free of glucose, glutamine and sodium pyruvate (Gibco, No. A14430). A549 cells cultured in the RPMI 1640 medium containing 11 mM glucose and 2 mM glutamine were used as a control. All treated and control A549 cells were disassociated with 0.25% Trypsin and resuspended in 1 mL DMEM medium (no glucose, no glutamine, no sodium pyruvate). After cell enumeration, 300,000 cells were isolated and incubated with 2-NBDG (400 μM) and C12R (1 μM) in the DMEM medium (no glucose, no glutamine, no sodium pyruvate) for 15 min, followed by processing with a BD LSRFortessa flow cytometry for fluorescence measurement.

To study the 2-NBDG and C12R response to metabolic inhibitors on cell lines, A549 cells were treated with 2-DG (5 mM), oligomycin (2 μM), phenoformin (25 μM), BPETS (10 μM), etomoxir (200 μM), and DMSO as the control for 12 h at 37 °C. Thirty minutes before assay, the media was changed to fresh media containing inhibitors at the same concentrations. 2-NBDG and C12R were then added to yield a concentration of 600 μM and 1 μM, respectively. In each condition, more than 5000 cells were assayed at the single-cell level.

To study the effect of metabolic inhibitors on metabolically active cells in pleural effusions, pleural effusion of Patient 30 was filtered, lysed of red blood cells and incubated with APC-conjugated anti-CD45 for 30 min. After washing and cell counting, 2-NBDG (400 μM) and C12R (1 μM) were used to fluorescently label metabolically active cells and applied onto 3% Matrigel-coated microwell chips at ~500,000 cells/chip via a 5-min rapid incubation. After on-chip washing with cold PBS and DMEM, the chips were sealed with porous membranes to avoid cell loss in the following steps. All cells on the chip were imaged and then treated with metabolic inhibitors (2-DG: 5 mM; phenoformin: 25 μM; BPETS: 10 μM; etomoxir: 200 μM) and DMSO as the control for 12 h at 37 °C. At the end of the treatment, PE Annexin V Apoptosis Detection Kit I was used to stain apoptotic cells on the chip followed by the second round of imaging. For data analysis, we counted the metabolically active cells identified by 2-NBDG and C12R, and then calculated the percentage of cell apoptosis in two metabolically active cell subpopulations (C12R$^{high}$ and 2-NBDG$^{high}$).

**ECAR and OCR measurements.** A549 cells were plated in the seahorse cell plate at 10,000/well and incubated overnight with RPMI 1640 medium supplemented with 10% FBS. A549 cells were then treated with inhibitors (2-DG: 5 mM; oligomycin: 2 μM; phenoformin: 25 μM; BPETS: 10 μM; etomoxir: 200 μM) and DMSO as a control in fresh complete cell culture media for 12 hours. Thirty minutes before assay, the media was changed to fresh media containing inhibitors at the same concentrations. ECAR and OCR were measured on an XFe96 Seahorse Biosciences Extracellular Flux Analyzer for 10 cycles. The averages of 10-cycle measurements of ECAR and OCR were recorded. Three replicates were performed in each condition.

**Preparation of single-cell suspensions from lung tissue.** Remnant tumor tissue samples obtained from bronchoscopy were immediately transported to the laboratory in the F12K/DMEM (1:1, Corning) medium, followed by enzymatic digestion with collagenase type I (170 mg L$^{-1}$, Gibco) and elastase (25 mg L$^{-1}$, Sigma-Aldrich) at 37 °C for 30–45 min. The dissociated cells were filtered with a 70 μm mesh cell strainer (BD) and centrifuged to a pellet at 300 × g for 10 min. After aspirating the supernatant, the cell pellet was resuspended in the red blood cells lysis buffer (BD Pham Lyse) and incubated for 2 min for lysing red blood cells. The cell pellet was washed twice with HBSS containing 0.1% BSA before performing on-chip metabolic assay.

**Patient survival analysis.** Survival was defined as the time between the start of malignant pleural effusions (MPE) collection until date of death or the last follow-up visit. The last date of follow-up was 10 January 2019. The study was approved by the institutional review board of the Shanghai Chest Hospital. The Kaplan–Meier method was used to estimate survival rate, along with a log-rank statistical test comparing the survival distribution. All tests were two sided, and p values <0.05 were considered statistically significant. The statistical analyses were performed with R software (version 3.3.3, R Foundation for Statistical Computing, Vienna, Austria) and RStudio software (version 1.1.383).

**Patient PET/CT scan.** FDG-PET/CT images were acquired on a Siemens Biograph mCT-S system using a standard protocol before invasive procedures were performed. Patients fasted 6 h and had a blood glucose level of <7.8 mmol/L prior to $^{18}$F-FDG administration. PET-CT scans were acquired 45–60 min after intravenous injection of 0.10–0.15 mCi kg$^{-1}$ (3.7–5.6 Mbq kg$^{-1}$) $^{18}$F-FDG. Patients were scanned in the supine position from the skull base to one-third of the femur (5–7 beds) at 2 min/bed. The tomographic images were reconstructed by using TrueX point spread function and time-of-flight iterative reconstruction algorithm. All patients underwent a 30-s breath-holding thin-slice CT scan. FDG-PET/CT images were independently reviewed by two experienced nuclear medicine physicians and

a final consensus was obtained on all imaging findings. Any accumulation of lesions outside the normal distribution area or above peripheral physiological uptake area was considered abnormal. The SUV was normalized by body weight, and the SUVmax was calculated as the highest value of the tumor voxel in each patient's primary lung tumor.

**PD-L1 immunofluorescence staining**. Pleural effusion of patient 31 was processed and assayed with APC-conjugated anti-CD45 antibody, 2-NBDG and C12R based on a protocol described above. After on-chip washing with cold PBS and DMEM, the chip was imaged and then sealed with a porous membrane. After on-chip cell fixation (2% PFA, 10 min) and blocking (3% BSA and 10% Normal Goat Serum), the chip was imaged again to confirm the disappearance of fluorescence of 2-NBDG and C12R. Cells on the chip were then stained with anti-PD-L1 (Biolynx, clone RR604) overnight at 4°C, and after extensive washing with PBS, cells treated with FITC-conjugated goat-anti-rabbit secondary antibody (Thermo Fisher Scientific) in PBS for 1 h.

**AXL inhibitor treatment on MPE samples**. Typically, 45 mL of pleural effusion was filtered by a membrane with a pore size around 100 μm, followed by lysis of red blood cells. The remaining nucleated cells were resuspended in HBSS and incubated with APC-conjugated anti-CD45 antibody for 30 min. After washing and cell counting, the cell suspension was applied onto three microwell chips and each chip was loaded with ~500,000 cells. The metabolic activities of these cells were assayed with 400 μM 2-NBDG and 1 μM C12R in glucose-free DMEM for 5 min at 37 °C. After imaging, the microwell chips were sealed with porous polycarbonate membranes (pore size: 3 μm) to avoid cell loss. The cells were then cultured on the chips and treated with DMSO vehicle, and R428 (1 μM) for 12 h at 37 °C, respectively. PE Annexin V Apoptosis Detection Kit I was used to measure the apoptosis of cells on the chips for calculating percentages of apoptosis in metabolically active subpopulations (C12R$^{high}$ and 2-NBDG$^{high}$).

**Partial least square discriminant analysis**. As the number of observations is low and the multicolinearity between measured variables is high, Partial Least Square Discriminant Analysis (PLS-DA) was employed to predict the membership of observations to the categories of responders and non-responders[40]. PLS-DA was performed with XLSTAT (Addinsoft) statistical software.

The goal of PLS-DA is to obtain a linear relationship between the measured variables and the patient response. PLS-DA begins with a matrix in which the number of columns is the number of measured variables and the number of rows is the number of patient samples (Supplementary Table 3). We seek a solution to that matrix that best resolves the responders from the non-responders with the most stable model. We used 29 patient data to construct the PLS-DA model. P6, P21, and P32 were excluded in the subsequent analyses due to lack of appropriate treatment, lack of determined diagnosis, and lack of follow-up information, respectively (Supplementary Table 1). Due to the relatively small sample size, we further categorized the partial response (PR) and stable disease (SD) as positive response (P), and progressive disease (PD) and death as negative response (N) (Supplementary Table 3). Herein, K is the number of categories (total 2, for either responder (P) or non-responder (N)) of the observation variable Y (the patient response). For each patient, we have 5 explanatory variables measured (Supplementary Table 3). For each category $a_k$ ($k = 1,2$), we obtain a separate classification function $F$, so that we obtain one fit that applies to all responders (P), and a second for all non-responders (N).

$$F(y_i, a_k) = b_0 + \sum_{j=1}^{p} b_i x_{ij} \qquad (1)$$

Here, $b_0$ is the fitted intercept of the linear model associated each category $a_k$, $p$ (total 5) is the number of measured explanatory variables and $b_i$ are the coefficients that weigh each variable within the model.

A given patient $i$ is associated to class $k$ (responder or non-responder) depending on which model best describes explanatory variables measured from that patient. An observation is assigned to the class with the highest classification function $F$. Formally, this is written as:

$$k^* = \arg\max_k F(y_i, a_k) \qquad (2)$$

Variable Importance for the Projection (VIP) can measure the importance of an explanatory variable for determining class membership of the observation variable $y$. The VIP for the $j$th explanatory variable is computed as

$$VIP_j = \frac{N \sum_{i=1}^{N} w_{ij}^2 RSS_i}{RSS_T} \qquad (3)$$

where $w_{ij}$ is a PLS-DA weight, $RSS_i$ is a percentage of the explained residual sum of squares, and $RSS_T$ a total percentage of the explained residual sum of squares[79].

**Statistical analysis**. Statistical analyses were performed using GraphPad PRISM 7 (GraphPad Software, Inc) and XLSTAT (Addinsoft, for PLS-DA modeling and correlation test) unless noted elsewhere. Statistical significance between two groups

were compared using two-tailed Student's $t$-test with $p < 0.05$ as the significance threshold. Alpha level was corrected by Bonferroni correction when multiple groups were compared.

**Reporting summary**. Further information on research design is available in the Nature Research Reporting Summary linked to this article.

## Data availability

Sequence data reported in this paper are available in the Sequence Read Archive (BioProject accession PRJNA554445).The data underlying all findings of this study are available within the article and its Supplementary Information files or from the corresponding authors upon reasonable request. A reporting summary and a source data file for this Article are available as a Supplementary Information files.

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

## Acknowledgements

We thank Dr. Bin-Bing S. Zhou for the critical comments and insightful discussion for this manuscript. We thank the following agencies and foundations for support: National Key R&D Program 2016YFC1303300 (to S.L.) and 2016YFC0900200 (to Q.S.); National Natural Science Foundation of China Grants 21775103 (to Q.S.) and 81672272 (to S.L.); Andy Hill CARE Fund (to W.W.); Washington Research Foundation Technology Development Grant (to W.W.); Phelps Family Foundation (to W.W.); Clinical Research Plan of SHDC 16CR3005A (to S.L.); Shanghai Chest Hospital Project of Collaborative

Innovative Grant YJXT20190209 (to S.L.). Z.L. is supported by Shanghai Science and Technology Commission Guidance Project 18411968200, Medical-Engineering Joint Funds of Shanghai Jiao Tong University YG2017MS81, Shanghai Chest Hospital Project of Collaborative Innovative Grant YJT20191015, Shanghai Youth Top Talent Project, and Shanghai Pujiang Talent Program (Grant No. 16PJD043). Y.L. is supported by National Natural Science Foundation of China Grants 81701852.

## Author contributions

S.L., W.W., and Q.S. designed and supervised the study. Z.L. and S.L. contributed clinical samples. Z.L., Z.W., Y.T., J.C., C.W., B.W., L.Y., and Z.G. performed the experiments. X.L. performed the bioinformatics analysis. Z.L., Y.T., X.L., Y.D., M.X., W.W., and Q.S. analyzed the data. W.W. and Q.S. wrote the paper.

## Additional information

**Competing interests:** The authors declare no competing interests.

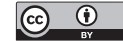

