## [Peer Review File · Nature Communications]

Reviewers' Comments:

Reviewer #1:

Remarks to the Author:

The Authors have presented an analysis of a single-cell imaging analysis platform and its application to malignant pleural effusions in lung adenocarcinoma patients. As the Authors point out, genetic changes such as EGFR mutations are not sufficient to predict the response of all patients to targeted therapy. And there are even fewer available markers available for prognostication or prediction of therapeutic response to cytotoxic chemotherapy. There has been recent focus on tumor cell metabolism, since variation in the glycolytic phenotype may explain cancer behaviour and treatment response. But there is no convenient way to measure tumor cell metabolism in the clinic, other than FDG-PET scans, which only provide information on glucose metabolism and not oxidative phosphorylation. Thus, the Authors set out to design and evaluate the potential clinical utility of a device that measures both glucose metabolism and oxidative phosphorylation within single cancer cells from malignant pleural effusion samples.

The findings in the submitted manuscript are of interest to clinical and translational researchers. The description of the assay development is detailed and informative. The advantages over their previously published assay that profiled glucose metabolism only are clearly described and justified. It is actually quite simple and elegant that so much information might be revealed by this 2-color cytometric assay that incorporates glucose metabolism and C12-resazurin, a mitochondrial redox indicator. This tool has the potential to have clinical utility in lung cancer patients, as the Authors describe in their manuscript. However, as described below, there are some shortcomings of the presented work that lead me to question the robustness of the assay, the validity of the Authors' conclusions, and ultimately the potential clinical utility of their device.

Specific Comments:

- 1) The technical platform appears to be well characterized. In some of the images there are doublets (i.e., 2 cells per well). How frequent does this occur and does this interfere with the analysis? Or perhaps these cells/wells can/should be excluded from analysis.
- 2) A major finding of this work is that the malignant pleural effusion cells behave very differently in this assay than the cancer cell lines. There is not enough exploration of the potential reasons for this, in the opinion of this reviewer. For instance, there is no mention or discussion of plasma membrane permeability. Dead cells would presumably not metabolize the C12R, but could 2-NBDG still be internalized and fluoresce? In the manuscript, the Authors suggest that dying cells would be in the 2-NBDG-low/C12R-low subset, but there is no specific analysis to support this. Could an over-abundance of dead cells in the pleural fluid explain the discrepancy between these samples and the cell lines (where there was high correlation between C12R and 2-NBDG signal)? Cells could die due to the harsh environment within the pleural fluid. Also could the RBC lysis buffer lead to tumor cell death? The RBC buffer was not used for the cell lines (understandably). Can the Authors use another method to evaluate cell viability and plasma membrane integrity to address this critical issue?
- 3) Could the RBC lysis lead to GSH release that could affect the C12R metabolism in the assay? Perhaps this also contributes to the difference between pleural fluid cells and cancer cell lines, since the cell lines were not treated with RBC lysis buffer.
- 4) The Authors appear to be speculating that cancer cells shed into the malignant pleural effusion fluid are representative of metastatic sites even outside of the chest. Is there actually any evidence for this? It does not appear that any evidence to this effect is presented in this manuscript.
- 5) The Authors mention that FDG-PET is already available in the clinic to measure tumor

metabolism (if only glucose uptake). Presumably some if not all of the patients in this study would have had FDG-PET. In light of this, it seems like a missed opportunity to not have compared their results to FDG-PET metrics of glucose metabolism. Is this data available?

6) Generalizability of these findings must be viewed with caution. Below are a few specific comments that touch on this:

— The Authors claim that their study applies to NSCLC, but really this is a study of adenocarcinoma. It is known that FDG uptake/metabolism differs between distinct lung cancer histologies (e.g., SCC vs adenocarcinoma), so the Authors should be careful regarding their conclusions. Adding another cohort of SCC would strengthen the study and could potentially allow for greater generalizability of their device.

— The small cohort size makes the prognostic analysis difficult to interpret. The Authors present multivariable and univariable analyses, and given the limited cohort I find the results compelling. The risk of overfitting is very high, though, so the Authors must be careful regarding their claims of generalizability to NSCLC (for example, non-adenocarcinoma types, as discussed above).

— The topic of immunotherapy is not brought up. This is relevant since immune checkpoint blockade is now standard of care for PD-L1-expressing metastatic NSCLC. The current cohort does not include patients treated with these agents. This could limit the generalizability and clinical relevance of the presented findings, as it is unknown whether the prognostic/predictive signals revealed by this device would apply to this setting. A comment or discussion regarding this point would be beneficial.

Minor Comment:

- Figures 2B&C are mislabeled in the figure legend.

Reviewer #2:

Remarks to the Author:

NCOMMS-18-28558

Li et al. "Pleural Effusion-Based Single-Cell Metabolic Phenotyping of Lung Cancer Patients for Informative Diagnosis"

In this manuscript, Li et al. utilize a metabolic on-chip assay to characterize tumor cells from NSCLC malignant pleural effusions based on markers of glycolytic vs mitochondrial oxidation. They find that these two states are largely mutually exclusive and that the presence of predominantly glycolytic cells correlates with poor patient performance irrespective of genotype or treatment. The glycolytic high cells exhibit a mesenchymal-like phenotype and are susceptible to AXL inhibition.

This study uses a novel methodologic approach (improved from a previous version of the assay that they previously described) to characterize tumor cells from patients to understand how different phenotypic metabolic states might affect tumor behavior and clinical outcome. This is an important and relevant question, because current treatment decisions in NSCLC are mostly driven by tumor genetics, yet patients with similar driver oncogenes may have variable responses to the same treatment. The authors should be commended for their innovative approach to addressing this question and for their provocative results suggesting that treatment outcome is more strongly determined by the relative composition of metabolic phenotypes of single tumor cells rather than the clinical characteristics typically used to make treatment decisions. My main concern, however, is that it is difficult to determine whether the data truly supports this conclusion based on the presentation of the clinical data. As the main goal of the manuscript is to correlate the metabolic profiling with clinical behavior, this is a major weakness that needs to be addressed.

Major points:

1. The clinical cohort in this study includes 26 NSCLC patients with malignant pleural effusions spanning different oncogenic subsets (EGFR, KRAS, ALK, ROS1, WT) and treatment histories (TKI, chemo, naive, etc.). While the authors have presented this as a strength of the study, the heterogeneity of the cohort and treatments make it challenging to judge the strength of the conclusion that metabolic phenotype is the primary determinant of outcome, especially since the outcome of the patients would be expected to vary depending on the specific prior and subsequent treatments patients receive. At face value, it seems surprising that metabolic phenotype would be a more important predictor of outcome than these factors that have been established by many clinical trials over many years. For instance, a wild-type patient would be expected to have much poorer response to chemotherapy in the 2nd vs 1st line setting. Similarly, an EGFR mutant patient who has progressed on EGFR TKI (with no T790M) would be expected to have a poor response to subsequent EGFR TKI. In addition, two different therapies may have significantly different outcomes in the same patient population – for instance, a patient with a ROS1 fusion (P16), would be expected to have different outcomes with crizotinib vs chemotherapy. Since the numbers of the cohort are relatively small, these factors could exert a rather large effect on the results. Thus, as presented, the clinical detail is not sufficient to be able to interpret the meaning of the response data. Additional clinical details that should be added or clarified are: (many of these could be added to Table 1)

- Which specific prior treatments did the patients receive - which exact TKIs, and for those listed as “chemo + targeted”, does this mean sequential therapies or concurrent? The authors should also list the clinical response to the prior therapy.
- For the reasons described above, the exact subsequent therapy or therapies after analysis of the pleural fluid cells needs to be reported for each patient.
- The manuscript would benefit from explicitly defining which patients were included in all the various analyses to be able to better interpret conclusions. For instance, in Figure 3, 23 patients out of the total 26 were included. Which patients were excluded and why? Which are the 7 patients who fall in the “good prognosis” group?
- It would be helpful if the ECOG performance scores and response class (PR, SD, PD) were also added to Table 1 so it is possible to know which patients and treatments are contributing to each group in the outcome analyses.
- The use of ECOG performance score (Fig 3A) is a bit unusual since this can be highly subjective and not an accurate measurement of the response of the tumor. Who performed this assessment and why was the time period of 5-7 months chosen evaluation? Also, at what time point were the RECIST measurements performed or were they “best response?” Were the assessments done in an independent and blinded manner? Did the pleural effusions respond consistent with the solid tumor lesions?

2. For the EGFR cohort, a number of questions are raised by the data. Similar to the comments above, clinical experience over the past several years has demonstrated different efficacy of TKIs and chemo in the context of different lines of treatment. From Figure 4A, it appears that almost all patients who had been previously treated with EGFR TKIs fall in the highly glycolytic or balanced groups and have SD or PD. Only one patient that previously received EGFR TKI fell in the mitochondrial oxidation/PR group. So, from this data, it appears difficult to say that the metabolic phenotype is independent from the already expected response to treatment. Did all the naive patients receive TKI, and if so, which TKIs? Did they receive subsequent treatment (chemo or other TKI after progressing on the first?). Did the patients who had become resistant to TKI receive subsequent chemo or TKI (for instance p17)? If a patient had already previously received chemo and TKI, what subsequent treatment did they then receive? One confounding factor in using SD as outcome is that patients will often begin to progress (the drug is no longer killing the cells) but meet criteria for stable disease – and often continued on post-progression TKI treatment – because the disease can be slow-growing.

3. The single cell assay is designed to identify metabolically active cells, with metabolically inactive cells being ignored. This assumes that all tumor cells will be metabolically active – however if there are quiescent cells (which have been well studied by many groups), then presumably these would be missed. Have the authors assessed the metabolically negative population to determine whether this contains quiescent tumor cells (viable, metabolically quiet, contain tumor mutations)? Conversely, there were ~20-30% of metabolically active cells that did not contain the oncogenic driver mutation. Were these mutation negative cells included or excluded from analysis? If they were included, can the authors demonstrate that these are indeed tumor cells that just appeared to be WT (perhaps based on limitations of the sequencing approach)? Is there any correlation between mutation-negative metabolically active cells and one metabolic phenotype or the other?

4. In Figure 6, the authors show that high glycolytic cells express AXL and can be targeted by AXL inhibition. As displayed, the fluorescent images are challenging to see (particularly AXL). It would be helpful if the authors could also display the data with quantification of number of AXL+ cells examined for each group (out of the total number examined), and if possible, the intensity quantification of each cell in each group. Also, it would be helpful to know how many patients were examined or if these cells are all from the same patient (and which specific patient(s) out of the cohort). Are these the same patients as in Figure 5? And in the AXLi treatment studies, which 3 patients are these?

5. The study is based on the use of two different fluorescent probes to identify highly glycolytic vs mitochondrial oxidation cells. As the authors correctly point out in the discussion, it is possible that tumor cells growing in a tissue microenvironment may have different metabolic profiles compared with tumor cells growing in suspension in a pleural effusion. In fact, they show that the two metabolic phenotypes are different in cells growing in culture, where the two phenotypes correlate rather than being mostly mutual exclusive. Thus, it seems important for the authors to show that these same phenotype distributions exist in tumor cell populations isolated from tumor tissue – this would be relatively straightforward to do from either mouse NSCLC xenograft tumors/PDXs or cell isolated from lung tumor surgical resection samples.

6. The rationale for the use of the fluorescent probes is that they identify high glycolytic or mitochondrial oxidation cells. The Enrichr scores suggest that the subset of DEGs shared by the 5 patients profiled are enriched for genes associated with ETC and oxidative phosphorylation, however it is not clear if this actually marks the C12R cells that are supposed to use mitochondrial oxidation (the GSEA results don't confirm this). Since the use metabolic phenotyping forms the underlying premise of the entire study, it would be helpful if the authors can provide some explanation for this apparent contradiction. Also, it would be useful to know if there are other gene signatures that are enriched in the C12R cells besides epithelial genes (Fig 5D-F)?

Minor points:

1. The figure legends for Figure 2 B and C are out of order.

2. The authors show that AXL inhibition induces apoptosis in glycolytic but not mitochondrial oxidation cells. It would be helpful to have a more detailed description of how the on-chip apoptosis assay was actually performed since this seems different from the more standard flow cytometry assay that most readers will be familiar with. In Figure 6B-C, what exactly do the Y axis units mean (I assume that a.u. = absorbance units)? Is the cell viability determined by counting the number of apoptotic cells? What is the actual frequency of apoptotic cells with AXLi and vehicle? The authors should provide the raw values for how many cells were counted, how many were scored as apoptotic, for both AXLi and baseline. In most applications looking at cells treated with RTK inhibitors, 12 hours is typically too short to see apoptosis occurring after drug treatment, so these details are essential for demonstrating the robustness of the assay.

3. In the discussion, the authors state (lines 395-398) "We speculate that a very similar mechanism may contribute to ..." The authors may wish to reference a recent study that showed that EGFR cells surviving EGFR TKI exhibited a mesenchymal phenotype, could evolve genetic mechanisms of acquired resistance TKI, and then were more tolerant to subsequent treatment with 3rd generation EGFR TKI. (PMID: 26828195)

4. In the discussion (lines 415 – 419), the authors state "Therefore our results suggest that NSCLC patients with high N/R ratios should probably take more aggressive treatments...or potentially resort to checkpoint immunotherapy..." As these studies are preliminary and have not even been tested using in vivo models, the authors may wish to refrain from making statements that appear to be clinical recommendations. In particular, clinical studies have shown that immune checkpoint inhibitors are largely ineffective in EGFR mutant patients, even when PDL1 expression is high (PMID: 29874546).

Reviewer #3:

Remarks to the Author:

The work by Li et al describes tumor cell heterogeneity measured by differential levels of glycolysis and mitochondrial oxidation. Using a very elegant technique to measure this at the single cell level they illustrate that the ratio between these two processes is predictive of tumor progression. The manuscript is very clearly written, the experiments are thoughtful and carefully done and the conclusions are measured and appropriate.

The one suggestion would be to revise the title of the paper to reflect what was measured was a ratio between glycolysis and oxidative phosphorylation. As written, one gets the impression that it is an overall metabolic profiling.

Reviewer #4:

Remarks to the Author:

Ziming Li et al. use an innovative new approach to profile the cells from malignant pleural effusion (MPE) using metabolic probes. Their work is novel because it provides a new method to profile MPEs during cancer treatment and potentially provides a new marker of response to current therapies for lung adenocarcinoma. Although the observations are very promising, the authors need to address some major concerns in relation to validating these new metabolic markers and demonstrating that they are relevant to non-EGFR mutant lung adenocarcinoma.

Major comments:

1) Underrepresentation of Kras-driven lung adenocarcinoma. The percent of NBDG^{high} cells in the one KRAS mutant patient 4 is high, which would agree with previous literature. However, additional patients with KRAS mutations would provide more proof of a genotype-specific difference in the glucose vs mitochondrial oxidation phenotype of MPEs. This is particularly important because most patients are EGFR mutant and therefore the conclusion the authors make about the N/R score in lines 238-241 about the importance of the N/R score in predicting response of patients with different genotypes is not supported by the data.

2) The authors should describe and graph the percentage of cells of each 4 populations in every patient. Based on the numbers for the CD45^{neg} cells in patient 1, it appears that very few cells of the total (~500k cells) are actually assayed. Based on Fig2D, this seems to be the case for most patients.

3) What is the explanation for some CD45^{neg} cells from patient 1 not having the EGFR ex19 deletion?

4) Why is there a discrepancy between MPE and cancer cell line data? The authors should

elaborate. Is this simply the in vivo vs in vitro growth conditions? When the authors culture cells for viability assays in Figure 6B-C, do they see any change in the N/R ratio?

5) How does the AUC of the ROC curve compare to other known markers of MPEs previously identified?

6) The authors present data that suggests that driver mutations, in particular EGFR mutations or CNVs (in 3 patients) are insufficient to explain the observed uncoupling of metabolic phenotypes in patients MPE samples. However instead of using those same 3 patients to see if gene expression can explain the metabolic phenotypes in the context of no oncogenic driver and CNV, they go on to analyze a random set of 5 patient MPEs by RNAseq. Are the gene expression differences in those 5 random MPE samples due to differences in oncogenic driver (eg Kras, EGFR) or CNVs?

Furthermore, additional mutations, independently of the known major oncogenic driver may be leading to metabolic rewiring of tumor cells. The authors should use their RNAseq data to call protein-altering mutations in the MPE samples. Furthermore, they should elaborate on the OXPHOS genes that are enriched in Fig5C. Given that they are able to stain cells from MPEs by immunofluorescence, they should check the levels of recurrently altered OXPHOS genes by IF.

7) The increased levels of AXL in the N/R high cells is interesting, particularly because of data in Fig6B-C showing a response of these cells to AXL inhibition. However, what remains unclear is whether AXL has a functional role in metabolic rewiring of cells, leading to an increase of the glycolysis over oxphos. Using the experimental setup in figure6 the authors should assess the following: 1) Does AXL inhibition lead to a decrease in the N/R ratio in 2-NBDGhigh/C12Rlow cells as compared to 2-NBDGlow/C12Rhigh cells? 2) the prediction would be that 2-NBDGhigh/C12Rlow cells would be more sensitive than 2-NBDGlow/C12Rhigh cells to metformin or other mitochondrial inhibitors. 3) Are 2-NBDGhigh/C12Rlow cells more sensitive to 2DG or other glycolytic inhibitors as compared to 2-NBDGlow/C12Rhigh cells. 4) Are 2-NBDGlow/C12Rhigh cells more reliant on alternative sources of carbon other than glucose to drive TCA cycle and OXPHOS? They should test whether 2-NBDGlow/C12Rhigh cells are more sensitive to inhibition of glutamine catabolism by glutaminase inhibition (eg BPTES, CB839).

8) Given that the authors are able to grow enough cells from MPEs to perform viability assays, it is critical that the authors use alternative methods to validate their N/R scores using a different platform. The use of seahorse would enable them to use low cell numbers and measure oxygen consumption and glycolysis (media acidification).

Minor comment:

Line 236 the authors should correct viable to variable.

We deeply appreciate all the reviewers for their insightful and constructive comments that significantly improved our manuscript. Please find below the point-by-point responses to reviewers' questions.

Reviewer #1

R1Q1: The technical platform appears to be well characterized. In some of the images there are doublets (i.e., 2 cells per well). How frequent does this occur and does this interfere with the analysis? Or perhaps these cells/wells can/should be excluded from analysis.

Response: There are a variety of cell types present in the pleural effusion with different sizes. The size of microwells (30 μm in diameter and 20 μm in depth) is thereby designed to accommodate large cells in the pleural effusion. For this reason, it is common to have multiple small cells in a single microwell, especially for those pleural effusions containing a large number of leukocytes. However, it does not interfere with cell imaging, analysis and single-cell retrieval. Only the cell-of-interest will be precisely retrieved by a motorized micromanipulator. The aim of using microwells to accommodate cells is to avoid cell loss during on-chip staining and washing, and importantly, to provide an address for each cell for subsequent single-cell manipulation. As shown in Fig. R1 below, the positions of cells in the microwells keep unchanged during cell imaging and subsequent single-cell retrieval, leading to precise manipulation of the target cell.

Figure R1. Precise single-cell retrieval from a microwell containing multiple cells. Left, before single cell retrieval; Right, after single cell retrieval.

Change: In the revised manuscript, we included Fig. R1 as a supplementary figure (Supplementary Fig. 1c) and added relevant discussion in the Result section to make this point clear.

R1Q2: A major finding of this work is that the malignant pleural effusion cells behave very differently in this assay than the cancer cell lines. There is not enough exploration of the potential reasons for this, in the opinion of this reviewer. For instance, there is no mention or discussion of plasma membrane permeability. Dead cells would presumably not metabolize the C12R, but could 2-NBDG still be internalized and fluoresce? In the manuscript, the Authors suggest that dying cells would be in the 2-NBDG-low/C12R-low subset, but there is no specific

analysis to support this. Could an over-abundance of dead cells in the pleural fluid explain the discrepancy between these samples and the cell lines (where there was high correlation between C12R and 2-NBDG signal)? Cells could die due to the harsh environment within the pleural fluid. Also could the RBC lysis buffer lead to tumor cell death? The RBC buffer was not used for the cell lines (understandably). Can the Authors use another method to evaluate cell viability and plasma membrane integrity to address this critical issue?

Response: We agree with the reviewer that cells could die in the pleural fluid because of the harsh environment. Dead cells would not metabolize the C12R, but 2-NBDG could be internalized due to the diffusion through the damaged cell membrane rather than glucose transporter-based uptake. In our previous PNAS paper (Tang, et al., *PNAS*, 2017, 114, 2544-2549), we used 2-NBDG, fluorescent anti-CD45, and a fluorescent dead cell maker EthD-1, to stain cells, followed by on-chip washing and imaging. We found that dead cells (EthD-1+ cells) showed a low unspecific background of 2-NBDG.

To address the reviewer's concern, we first used A549 cell line to investigate the unspecific 2-NBDG signal of dead cells. A549 cells were exposed to hypotonic distilled water for 3 min followed by restitution in HBSS for 2-NBDG staining and washing. Meanwhile, viable A549 cells and leukocytes were assayed with 2-NBDG in parallel. Fig. R2 shows the fluorescent images of a typical viable A549 cell, a viable leukocyte and a dead A549 cell that is identified by CD45/2-NBDG/EthD-1 staining. Figure R3 further shows the 2-NBDG uptake of viable leukocytes (EthD-1-/CD45+), viable A549 (EthD-1-/CD45-) and dead A549 cells (EthD-1+/CD45-). Obviously, viable A549 cells exhibit a significantly higher 2-NBDG uptake level than that of dead A549 cells. For this reason, dead tumor cells present in the pleural effusion samples do not interfere with both 2-NBDG and C12R staining, and would not change the N/R ratios measured from pleural effusion samples in this study.

Figure R2. Fluorescent images of a viable A549 cell, a viable leukocyte and a dead A549 cell.

Figure R3. 2-NBDG uptake of viable leukocytes, viable A549, and dead A549 cells.

To evaluate the effect of RBC lysis buffer on the tumor cell viability, we performed RBC lysis protocol on A549 cell line including the RBC lysis buffer treatment, centrifuging, and washing. A Live/Dead Viability/Cytotoxicity kit (Thermo Fisher, Catalog #L3224) was employed to evaluate viability of untreated A549 cells and A549 cells treated with RBC lysis buffer. Each condition has three replicates with each replicate containing measurement of more than 100,000 cells. As shown in Fig. R4, there is no statistical significance between two groups. Therefore, RBC lysis buffer does not affect cell viability of tumor cells.

Figure R4. Viability of untreated A549 cells (control) and A549 cells treated with RBC lysis (RBC lysis). NS, not significant.

An orthogonal method was used to further validate the point. We took 5 mL of pleural effusion (PE) from Patient #30 and split it into two aliquots with 2.5 mL of each. The first aliquot of the

PE sample was directly analyzed following the protocol described in the paper. For the second aliquot, MACS Miltenyi Biotec Dead Cell Removal Kit was employed to remove dead cells before analyzing the 2-NBDG and C12R in the same way as the first aliquot. The N/R ratios were calculated to be 4.6 (156 CD45^{neg}/2-NBDG^{high}/C12R^{low} cells vs 34 CD45^{neg}/2-NBDG^{low}/C12R^{high} cells) and 4.4 (172 CD45^{neg}/2-NBDG^{high}/C12R^{low} cells vs 39 CD45^{neg}/2-NBDG^{low}/C12R^{high} cells) for the first and second aliquot, respectively. No significant difference was observed between two methods, indicating that dead cells did not interfere with the N/R ratio determination.

Change: In the revised manuscript, we included Figs. R2 and R3 as supplementary figures (Supplementary Fig. 1a, b) and added relevant discussion in the Result section to clarify the point.

R1Q3: Could the RBC lysis lead to GSH release that could affect the C12R metabolism in the assay? Perhaps this also contributes to the difference between pleural fluid cells and cancer cell lines, since the cell lines were not treated with RBC lysis buffer.

Response: We evaluated the C12R reaction kinetics with common intracellular reducing agents such as glutathione (GSH), ascorbic acid, glucose, and glutamine in our original manuscript. As shown in Fig. 1d (i.e. Fig. R5 below), we found that those reductants at biologically-relevant levels led to insignificant fluorescence increase of the C12R signal within 1 hour assay duration. The incubation time of C12R in the OMC assay was only 15 minutes (much less than 1 hour). NADH/NAD⁺ had the fastest reaction kinetics, and the reduction of C12R was primarily attributed to different oxidoreductase enzyme systems that use NAD(P)H as the primary electron donor.

Figure R5. Comparison of the C12R reaction using different reducing agents. Specifically, 100

μL of reducing agent (Glutamine 4 mM, Glutathione 16 mM, Glucose 12 mM, ascorbate 1.6 mM, NADH 1.2 mM in 10 mM PBS buffer) was mixed with 100 μL of diaphorase and 200 μL of C12R (2 μM in 10 mM PBS buffer), respectively, and incubated for 1 hour. The fluorescence intensity was determined by a microplate reader (540 nm excitation and 590 nm emission). Then the experiment was carried out in quadruplets to determine the error range.

R1Q4: The Authors appear to be speculating that cancer cells shed into the malignant pleural effusion fluid are representative of metastatic sites even outside of the chest. Is there actually any evidence for this? It does not appear that any evidence to this effect is presented in this manuscript.

Response: Malignant pleural effusion (MPE) represents advanced malignant disease. Median survival time following a diagnosis of MPE usually ranges from 4 to 12 months. In particular, lung cancer patients with MPE have the shortest survival time. For this reason, the recently revised 7th edition of the tumor-node-metastasis (TNM) staging system for non-small cell lung cancer (NSCLC) upstaged the presence of MPE from T4 to M1a by the International Association for the Study of Lung Cancer (Goldstraw P, et al. *J. Thorac. Oncol.* 2007, 2, 706–714.). NSCLC patients with MPE have similar survival time and treatment strategies with patients that have distant metastases, indicating high propensity for seeding distant metastases of tumor cells present in MPE. Bai et al. characterized the copy number variation (CNV) patterns of circulating tumor cells (CTCs), primary and metastatic tumors of lung adenocarcinoma patients, and found that those CTCs exhibited reproducible CNV patterns similar to those of the metastatic tumor of the same patient (see: Bai F, et al. *PNAS* 2013, 110, 21083–21088).

Change: We added these references to the relevant discussion parts in the revised manuscript.

R1Q5: The Authors mention that FDG-PET is already available in the clinic to measure tumor metabolism (if only glucose uptake). Presumably some if not all of the patients in this study would have had FDG-PET. In light of this, it seems like a missed opportunity to not have compared their results to FDG-PET metrics of glucose metabolism. Is this data available?

Response: In principle, 2-NBDG-based fluorescent glucose uptake assay should deliver similar information to PET imaging that utilizes a radioactive glucose analog, ¹⁸F-fluoro-2-deoxy-D-glucose (FDG), to measure glucose uptake. In our previous PNAS paper, we have shown that 2-NBDG is consistent with FDG in quantifying *in vitro* glucose uptake of cells (See Figure S3 in Tang et al., *PNAS* 2017, 114, 2544-2549). PET imaging measures glucose uptake of cells in tumors and neighboring tissues with a low spatial resolution *in vivo*, whereas 2-NBDG assay measures glucose uptake of cells in pleural effusions *ex vivo* at the single-cell resolution. PET imaging reports the maximum standardized uptake value (SUVmax) of the tumor and a SUVmax threshold of 2.5 has been widely applied in clinic to differentiate benign from malignant lesions (Hain *et al.*, *Eur J Nucl Med.*, 2001, 28, 1336-1340). Thus, SUVmax determined by the PET imaging represents the population of tumor cells with highest FDG uptake *in vivo*, and these cells may be comparable to the cell population with high 2-NBDG uptake in pleural effusions.

Our patient samples were collected in China where PET imaging is costly and not reimbursed. Therefore, only a small percentage of patients have FDG-PET data available. In addition, even the patients had PET imaging performed, the time of imaging was usually not concurrent with the pleural effusion measurement. In our dataset, only three patients (P15, P25 and P30) conducted PET imaging concurrently with MPE analysis. The SUVmax values of P15, P25, and P30 were reported to be 22.0, 19.6 and 7.4, respectively. Meanwhile, the 2-NBDG uptake of the CD45^{neg}/2-NBDG^{high} cells of these three patients are drawn in Fig. R6. The 2-NBDG signals have been normalized to the their respective cut-off values determined by 2-NBDG uptake of leukocytes (mean + 5×SD). The averaged normalized 2-NBDG uptake values across the three patients have consistent trend with their SUVmax values of PET imaging (Table R1 below).

However, FDG-PET only evaluates the glucose uptake the tumor tissue. Therefore, compared with the metabolic phenotyping via N/R ratios, it may be less predictive for patient therapy responses. For example, all the three patients evaluated here are newly diagnosed (Supplementary Table 1). The N/R ratio of P15, P25, P30 is 0.35, 1.18, and 4.70 respectively (Fig. R6). Although P15 has the highest SUVmax in PET imaging and the highest relative 2-NBDG intensity of 2-NBDG^{high} cell population in the pleural effusion among the three patients, this patient has more C12^{high} tumor cells present in the pleural effusion and consequently a lower N/R ratio and better response and survival (Table R1, see also Table 1 in the main text). In contrast, P30 with lowest SUVmax but highest N/R ratio has poorest response and shortest survival. The results indicate that our metabolic phenotyping assay could potentially provide complementary information to PET imaging for more informative cancer diagnostics.

Figure R6. Left, Maximum-intensity-projection PET images (left), axial CT (middle) and fused PET/CT images (right) showed variable FDG uptake in the pulmonary lesions across the three patients. Right, Normalized 2-NBDG uptake of 2-NBDG^{high} tumor cells in MPE samples overlaid with the SUVmax values from the PET images of the three patients. The 2-NBDG signals have been normalized to the their respective cut-offs determined by 2-NBDG uptake of leukocytes for each patient.

Table R1. List of patient SUVmax, 2-NBDG uptake, N/R ratio, therapy response, and survival.

	P15	P25	P30
SUVmax of PET	22.0	19.6	7.4
Normalized 2-NBDG uptake	2.60 ± 0.94	2.08 ± 0.53	1.61 ± 0.70
N/R ratio	0.35	1.18	4.70
Survival after PET imaging & MPE analysis	15.9 months (alive)	14.3 months (alive)	3.6 months

Change: We included Fig. R6 as part of the new Fig. 4. We also added relevant analysis and elaboration in the Result section to highlight that our metabolic phenotyping assay can provide information complementary to clinical FDG-PET imaging for more informative cancer diagnostics.

R1Q6: Generalizability of these findings must be viewed with caution. Below are a few specific comments that touch on this:

— The Authors claim that their study applies to NSCLC, but really this is a study of adenocarcinoma. It is known that FDG uptake/metabolism differs between distinct lung cancer histologies (e.g., SCC vs adenocarcinoma), so the Authors should be careful regarding their conclusions. Adding another cohort of SCC would strengthen the study and could potentially allow for greater generalizability of their device.

Response and Change: We agree with the reviewer that FDG uptake/metabolism differs between lung adenocarcinoma (ADC) and lung squamous cell carcinoma (SCC). Goodwin *et al.* found a critical reliance on glycolysis of lung SCC compared with lung ADC and that elevated GLUT1-mediated glycolysis in lung SCC strongly correlates with high ¹⁸F-FDG uptake and poor prognosis (Goodwin J, et al. *Nat. Commun.* 2017, 8, 15503). Compared with lung ADC, lung SCC rarely sheds cells into an effusion. In a retrospective study conducted in North India, only 2.7% of malignant pleural effusions contained atypical squamous cells and 77% of cases revealed malignancy of ADC (Awasthi A. et al. *Cytopathol.* 2007, 18, 28–32). In another retrospective study of lung cancer patients with MPE in South Korea (Lee et al. *BMC Cancer*, 2017, 17, 557), 85% of cases were of lung ADC and only 5.7% of cases were of lung SCC. In our study, we didn't rule out MPE samples of lung SCC patients during our sample collection, but failed to obtain MPE samples of SCC patients. For this reason, it is challenging to add another cohort of SCC patients. As suggested by the reviewer, we revised the text to confine the scope of our conclusions to lung ADC.

— The small cohort size makes the prognostic analysis difficult to interpret. The Authors present multivariable and univariable analyses, and given the limited cohort I find the results compelling. The risk of overfitting is very high, though, so the Authors must be careful regarding their claims of generalizability to NSCLC (for example, non-adenocarcinoma types, as discussed

above).

Response and Change: We appreciate that our cohort size is relatively small. In the revised manuscript, we added six more patients (now the total cohort size is 32) and re-performed the various analyses made in the original manuscripts. We found that our conclusions still hold in the expanded cohort size. Unfortunately, due to the reasons stated in R1Q6, we did not obtain MPE samples of SCC patients. We therefore limited our conclusions to lung adenocarcinoma as suggested by the reviewer. For the multivariate PLS-DA analysis, we evaluated the model quality and performed leave-one-out cross validation (Supplementary Table. 4) to ensure the model is not over fitted. Additionally, the predictive capability of the N/R ratio for patient prognosis has also been validated through correlation analyses in Fig. 3a, segregation analysis in Fig. 4a,b, and survival analysis in Fig. 4c. All the results are statistically significant and support the predictive capability of metabolic phenotyping.

— The topic of immunotherapy is not brought up. This is relevant since immune checkpoint blockade is now standard of care for PD-L1-expressing metastatic NSCLC. The current cohort does not include patients treated with these agents. This could limit the generalizability and clinical relevance of the presented findings, as it is unknown whether the prognostic/predictive signals revealed by this device would apply to this setting. A comment or discussion regarding this point would be beneficial.

Response: The reviewer raised an important point of immunotherapy. In our data, we indeed observed that the 2-NBDG^{high} cells are having higher checkpoint ligand expression levels (PD-L1, PD-L2, etc.) than C12R^{high} cells (Fig. 5f and Supplementary Fig. 13). This points to a possibility that patients with high N/R ratios may have higher PD-L1/PD-L2 expression in their tumor tissues. Given the patients with high N/R ratios are unlikely to get benefit from traditional chemo- or targeted therapies, alternative treatment strategies, such as first line checkpoint immunotherapy, might be considered for this poor-prognosis population. Of course, this hypothesis needs to be rigorously validated in a large patient cohort, which will be pursued in subsequent studies. Additionally, our patient samples were collected in China where Opdivo was just approved for second line treatment of NSCLC on June 15th, 2018 and the cost is high. Keytruda was approved for melanoma treatment on July 25th, 2018 in China but has not been approved for lung cancer treatment. Therefore, lung cancer patient samples treated with checkpoint immunotherapies are very limited and not available for this study.

Change: We added a discussion regarding this point in the revised manuscript. We pointed out that the conclusions cannot be generalized to predict responses to checkpoint immunotherapies. The robustness and generality of the predictive capability of the N/R ratio in lung cancer requires further examination in a larger patient population, particularly for patients receiving checkpoint inhibitors.

Minor Comment:

R1Q7: Figures 2B&C are mislabeled in the figure legend.

Response and Change: We have corrected this mislabeling.

Reviewer #2

R2Q1: The clinical cohort in this study includes 26 NSCLC patients with malignant pleural effusions spanning different oncogenic subsets (EGFR, KRAS, ALK, ROS1, WT) and treatment histories (TKI, chemo, naïve, etc.). While the authors have presented this as a strength of the study, the heterogeneity of the cohort and treatments make it challenging to judge the strength of the conclusion that metabolic phenotype is the primary determinant of outcome, especially since the outcome of the patients would be expected to vary depending on the specific prior and subsequent treatments patients receive. At face value, it seems surprising that metabolic phenotype would be a more important predictor of outcome than these factors that have been established by many clinical trials over many years. For instance, a wild-type patient would be expected to have much poorer response to chemotherapy in the 2nd vs 1st line setting. Similarly, an EGFR mutant patient who has progressed on EGFR TKI (with no T790M) would be expected to have a poor response to subsequent EGFR TKI. In addition, two different therapies may have significantly different outcomes in the same patient population – for instance, a patient with a ROS1 fusion (P16), would be expected to have different outcomes with crizotinib vs chemotherapy. Since the numbers of the cohort are relatively small, these factors could exert a rather large effect on the results. Thus, as presented, the clinical detail is not sufficient to be able to interpret the meaning of the response data. Additional clinical details that should be added or clarified are: (many of these could be added to Table 1)

Response and Change: We agree with the reviewer that the outcomes of the patients would vary depending on the specific prior and subsequent treatments received as well as various other clinical factors that have been established by clinical trials over years. In our patient cohort, we also observed wild-type or *KRAS*-mutant patients having poorer response to chemotherapy in the 2nd vs 1st line setting. For example, patient #4 bearing *KRAS*^{G12C} mutation had a much poorer response to the 2nd line chemotherapy than 1st line chemotherapy. In the meantime, the metabolic phenotyping of his pleural effusion drawn during the 2nd line chemotherapy showed a high N/R ratio (N/R ratio = 6.94). Therefore, in this scenario, the result of our metabolic phenotyping assay is wholly consistent with the various clinical experiences. We would like to clarify that we do not claim that the metabolic phenotyping is superior to other clinical factors (genetics, pathology, physiology, etc.). Instead, we believe that the metabolic phenotyping can provide information complementary to tumor genetics and other clinical factors for an improved cancer diagnostics. For example, tumor genetics can identify whether the patients are bearing targetable driver oncogene mutations and thus segregate patients to various chemo- and targeted therapy regimens. The metabolic phenotyping can further reveal whether the patients are likely (or unlikely) to benefit from the standard chemo- or targeted therapies identified by tumor genetics. This is important particularly for newly diagnosed patients who may benefit from such predictions prior to the onset of therapy.

In our expanded patient cohort in the revised manuscript, we have 14 newly diagnosed patients from whom we analyzed the MPE samples prior to the onset of the 1st line therapy. All the 14

patients received appropriate anti-tumor treatment tailored to their own disease status (WT: chemo; *KRAS* mutant: chemo; *EGFR* mutant: EGFR-TKI; *ALK*: crizotinib, etc.). As shown in Fig. 4a,c of the revised manuscript, the metabolic phenotyping accurately predicted their clinical responses and survival (purple dots in Fig. 4a, middle panel of Fig. 4c). Among the 14 newly diagnosed patients, 3 patients who were having mitochondrial oxidation phenotype ($N/R < 0.5$) had partial response and are still alive; 5 patients who were having glycolysis phenotype ($N/R > 2$) were all dead within 6 months before the follow-up; In the remaining 6 patients with a balanced phenotype ($0.5 < N/R < 2$), 5 out of the 6 were having a stable disease upon the follow-up. P16 bearing *ROSI* gene alteration was receiving the 1st line treatment before the approval of crizotinib by China FDA (September 2017). He therefore was only able to receive chemotherapy at that time.

The metabolic phenotyping can also predict varied therapy responses for patients with the same genotype and similar treatment history. For instance, Patients #1, #8 and #9 were all bearing *EGFR*^{19Del} and receiving EGFR-TKI (gefitinib) as a 1st line treatment. Consistent with our prediction, patient #1 with a low N/R ratio was having a PR upon follow-up and is still alive. Patients #8 and #9 with high N/R ratios were dead before the follow-up (Supplementary Table 1).

Taken together, these results demonstrate the unique value of the metabolic phenotyping for predicting patient therapy responses to the standard clinical treatments determined by the tumor genetics and other clinical factors. However, the robustness of the predictive capability of the metabolic phenotyping assay needs to be further validated prospectively in a larger patient cohort before translating to the clinic. This pilot study sets a stage for such a large prospective clinical validation.

- Which specific prior treatments did the patients receive - which exact TKIs, and for those listed as “chemo + targeted”, does this mean sequential therapies or concurrent? The authors should also list the clinical response to the prior therapy.

We have listed relevant information in the Supplementary Table 1 of the revised manuscript. All the patients were evaluated as PD to the prior therapies before proceeding to the current treatment.

- For the reasons described above, the exact subsequent therapy or therapies after analysis of the pleural fluid cells needs to be reported for each patient.

We have listed relevant information in the Supplementary Table 1.

- The manuscript would benefit from explicitly defining which patients were included in all the various analyses to be able to better interpret conclusions. For instance, in Figure 3, 23 patients out of the total 26 were included. Which patients were excluded and why? Which are the 7 patients who fall in the “good prognosis” group?

We listed the inclusion/exclusion information as well as prognosis information in the Supplementary Tables 1 and 3, and revised the main text to clarify the inclusion/exclusion

criteria. Briefly, in our patient cohort, we excluded P6, P21 and P32 from all the correlation analysis, PLS modeling, and survival analysis. It is because that P6 was not receiving appropriate anti-tumor treatment after diagnosis due to financial reasons. He instead only received palliative care with traditional Chinese medicine. P21 had an underdetermined pathological diagnosis and thus did not receive any clinical treatment. P32 was a most recent patient analyzed whose 1st line treatment has not started yet. So, we do not have any subsequent follow-up information. All the information above was noted in the Supplementary Table 1.

- It would be helpful if the ECOG performance scores and response class (PR, SD, PD) were also added to Table 1 so it is possible to know which patients and treatments are contributing to each group in the outcome analyses.

Due to the space limitation, we included all the therapy information, ECOG scores and RECIST responses in the Supplementary Table 1.

- The use of ECOG performance score (Fig 3A) is a bit unusual since this can be highly subjective and not an accurate measurement of the response of the tumor. Who performed this assessment and why was the time period of 5-7 months chosen evaluation? Also, at what time point were the RECIST measurements performed or were they “best response?” Were the assessments done in an independent and blinded manner? Did the pleural effusions respond consistent with the solid tumor lesions?

The ECOG score evaluation was strictly adherent to the ECOG formula and was performed in an independent and blinded manner. The RECIST measurements used in this paper were performed at around 6 months after MPE diagnosis and by board-certified radiologists and radiation oncologists in an independent and blinded manner.

The reason for choosing 6 months as the time period for re-evaluation is based on the evidence that lung cancer patients with MPEs survived for a shorter period (5.49 months) than those without pleural effusions (12.65 months) (Porcel J. M., *Curr. Opin. Plum. Med.* 2016, 22, 356-361; Porcel J. M., et al. *Respirology*, 2015, 20, 654-659). Therefore, we chose the 6-month that is close to reported average survival (5.49 months) as a follow-up time point for review of patient therapy responses (via RECIST) in this study. However, some patients failed to visit hospital precisely at 6 months. We therefore have 5-7 months' time period. It should be noted that physicians follow up patients according to clinical practice guidelines and protocols. The 5-7 month RECIST evaluation results were particularly selected for this study for the reasons stated above. We have revised the text to clarify these details.

The pleural effusions respond fairly consistent with the solid tumor lesions. The pleural effusions were not present at the follow-up for all the patients with PR and SD status. However, the pleural effusions were recurrent at the follow-up for 6 out of the 17 patients with PD or death status.

R2Q2: For the EGFR cohort, a number of questions are raised by the data. Similar to the comments above, clinical experience over the past several years has demonstrated different efficacy of TKIs and chemo in the context of different lines of treatment. From Figure 4A, it

appears that almost all patients who had been previously treated with EGFR TKIs fall in the highly glycolytic or balanced groups and have SD or PD. Only one patient that previously received EGFR TKI fell in the mitochondrial oxidation/PR group. So, from this data, it appears difficult to say that the metabolic phenotype is independent from the already expected response to treatment. Did all the naïve patients receive TKI, and if so, which TKIs? Did they receive subsequent treatment (chemo or other TKI after progressing on the first?). Did the patients who had become resistant to TKI receive subsequent chemo or TKI (for instance p17)? If a patient had already previously received chemo and TKI, what subsequent treatment did they then receive? One confounding factor in using SD as outcome is that patients will often begin to progress (the drug is no longer killing the cells) but meet criteria for stable disease – and often continued on post-progression TKI treatment – because the disease can be slow-growing.

Response and Change: This question is related to the previous comment that “an EGFR mutant patient who has progressed on EGFR TKI (with no T790M) would be expected to have a poor response to subsequent EGFR TKI”. We apologize for the confusion that we did not separate the patients with previous treatments from patients whose MPE samples were analyzed shortly after the start of the 1st line treatment. We listed both cases as patients with prior treatments in the original Table 1. Now we updated the information to include all the detailed treatment information in the new Supplementary Table 1.

In the updated Supplementary Table 1, only two patients (P5 and P26) bearing *EGFR*-mutant tumors were treated with and developed resistance to the first generation EGFR TKI (Gefitinib) therapies. We performed metabolic phenotyping on the MPE samples collected from these two patients before (for P26) or shortly after (for P5) they were treated with subsequent 3rd generation EGFR-TKI (Osimertinib). As we discussed in R2Q1, in this scenario, the result of our metabolic phenotyping assay is consistent with the various clinical experiences. For example, P5 with N/R>1 had a progressive disease and poor prognosis, which is in line with the clinical observation that “an EGFR mutant patient who has progressed on EGFR TKI (with no T790M) would be expected to have a poor response to subsequent EGFR TKI” as mentioned by the reviewer. On the other side, P26 developed resistance to Gefitinib due to the emergence of secondary *EGFR*^{T790M} mutation. Therefore, his tumor progression can be partially controlled by Osimertinib and got a SD status upon follow-up.

In addition to yielding information consistent with established clinical experiences, metabolic phenotyping assay can deliver new/predictive information that are not revealed by other clinical factors. As we already discussed in R2Q1, in the 14 newly diagnosed patients whose MPE samples were analyzed prior to the onset of therapy as well as the 5 additional patients (P7, P8, P17, P27, P29) whose MPE samples were analyzed shortly after the onset of the 1st line treatment and well before the RECIST evaluation, the metabolic phenotyping made fairly accurate prediction on the patient responses to the subsequent (or ongoing) therapies (Fig. 4a). These results highlight the predictive capability of the metabolic phenotyping assay that may deliver information complementary to other established clinical factors for improved cancer diagnostics for lung adenocarcinoma patients.

Regarding the comment on the confounding aspect of using SD as outcome, we appreciate the reviewer's point and it indeed sometimes happens in the clinic. We therefore went back to check the clinical records of the 4 patients (P2, P18, P19, P26) in the EGFR cohort with an SD outcome. All the 4 patients are still alive and we found that they had no therapy change and disease progression at least a few months after the RECIST evaluation as SD. Therefore, we believe that the SD status is an accurate evaluation in these cases.

R2Q3. The single cell assay is designed to identify metabolically active cells, with metabolically inactive cells being ignored. This assumes that all tumor cells will be metabolically active – however if there are quiescent cells (which have been well studied by many groups), then presumably these would be missed. Have the authors assessed the metabolically negative population to determine whether this contains quiescent tumor cells (viable, metabolically quiet, contain tumor mutations)? Conversely, there were ~20-30% of metabolically active cells that did not contain the oncogenic driver mutation. Were these mutation negative cells included or excluded from analysis? If they were included, can the authors demonstrate that these are indeed tumor cells that just appeared to be WT (perhaps based on limitations of the sequencing approach)? Is there any correlation between mutation-negative metabolically active cells and one metabolic phenotype or the other?

Response and Change: We agree with the reviewer that tumor cells are not all metabolically active. Quiescent and apoptotic tumor cells, as well as other cell types present in the pleural effusion such as mesothelial cells and leukocytes, are often metabolically inactive. We set up a high cut-off value of the metabolic activity to determine metabolically active tumor cells for two reasons. First, these cells represent a viable and pure tumor cell population that has been validated by single-cell sequencing. Second, these metabolic outliers contain unique information, and metabolic phenotyping of these cells enables establishment of an easy and rapid assay with clinical significance shown in the manuscript.

As shown in Supplementary Fig. 5, more than 70% of metabolically active cells were harboring driver mutations that were consistent with primary tumors. In the Fig. R7 below, 16 metabolically active cells from P1 were isolated, followed by single-cell genome amplification and Sanger sequencing. Thirteen of them (~81%) were found to harbor *EGFR*^{19Del} and the remaining three cells exhibited wild type *EGFR* (one 2-NBDG^{high}/C12R^{low} and two 2-NBDG^{low}/C12R^{high}). To determine the malignancy of the three *EGFR*^{WT} cells, we performed whole-genome sequencing (~0.1× sequencing depth) of all sixteen metabolically active cells and three white blood cells to characterize single-cell copy number (CNV) patterns (Fig. R7). CNV allows us to survey the entire genomic landscape and determine the malignancy based on chromosomal gains and losses. Two *EGFR*^{WT} cells exhibit gain and loss CNV patterns consistent with other *EGFR*^{19Del} cells, which can be identified as tumor cells. While one *EGFR*^{WT} cell shows distinct global CNV profile from other cells, its dramatic copy number alteration across many chromosomal regions (compared to WBCs) suggests a high likelihood of malignancy. It is presumably derived from a different colony of the primary tumor due to the intratumoral heterogeneity.

Additionally, in our previous PNAS paper (Tang et al., *PNAS* 2017, 114, 2544-2549), we performed the whole exome sequencing (WES) on $KRAS^{WT}$ cells isolated from the MPE sample of a $KRAS$ -mutant LADC patient. We screened the mutations with the Qiagen's Lung Cancer Panel, containing 45 most relevant driver oncogenes and tumor suppressor genes in lung cancer. These $KRAS^{WT}$ cells contained alterations in several marker oncogenes and tumor suppressors, including *BRAF*, *EGFR*, *PIK3CA*, *PTEN*, and *TP53*. These data also suggest the malignancy involvement of the $KRAS^{WT}$ cells.

In addition to the data shown above, we performed CNV analyses of patient MPE samples in many pilot studies during the development of the on-chip metabolic cytometry platform. The accumulated results suggest that most of the WT/CD45^{neg} metabolically active cells are indeed tumor cells with CNV patterns consistent to the cells bearing driver oncogene mutations. We therefore include all the CD45^{neg} metabolically active cells in the analysis. The cells with wild type driver oncogene may be derived from intratumor heterogeneity or be attributed to the limitations of single-cell genome amplification technique. Since only a small percentage of metabolically active cells exhibiting wide type driver oncogenes without metabolic phenotype preference, using detected mutant metabolically active cells in replace of all metabolically active cells to calculate N/R ratios does not change the main conclusions of this paper. We have added Fig. R6 as Supplementary Fig.6 and relevant discussion in the revised manuscript.

We indeed assessed the metabolically negative population. It contains much less tumor cells compared with metabolically active population. Given the relatively large number of metabolically negative cells which normally contain dying cells and other confounding cell types, as well as no gold standard markers for staining quiescent tumor cells, it is technically challenging to examine all the metabolically negative cells to identify quiescent tumor cells and distinguish them from stressful/dying tumor cells by single-cell genomic sequencing.

Figure R7. The single-cell CNV profiles and detected *EGFR* mutation status of metabolically active cells and white blood cells from P1.

R2Q4. In Figure 6, the authors show that high glycolytic cells express AXL and can be targeted by AXL inhibition. As displayed, the fluorescent images are challenging to see (particularly AXL). It would be helpful if the authors could also display the data with quantification of number of AXL+ cells examined for each group (out of the total number examined), and if possible, the intensity quantification of each cell in each group. Also, it would be helpful to know how many patients were examined or if these cells are all from the same patient (and which specific patient(s) out of the cohort). Are these the same patients as in Figure 5? And in the AXLi treatment studies, which 3 patients are these?

Response and Changes: We adjusted the brightness and contrast of Fig. 6a for better presentation of AXL levels. Meanwhile, AXL intensity quantification (with cell # assayed) of three metabolic subpopulations (2-NBDG^{high}, C12R^{high}, and double negative) from pleural effusion of P29 was shown in Fig. R8 below and included as Supplementary Fig. 14 in the revised manuscript. P29 was the patient for evaluating AXL expression in Fig. 6a. This patient was not included in the patient cohort of the initial submission because this patient did not reach the follow-up time point at the time of submission. However, the patient is now included in the revised manuscript with follow-up evaluation and survival analysis. While P29 was not one of the patients for RNA-seq analysis in Fig. 5, the RNA-seq results also showed consistent higher *AXL* gene expression levels in 2-NBDG^{high} cells than in C12R^{high} cells across other 5 patients analyzed (Fig. 5f). Therefore, we totally examined 6 patients whose 2-NBDG^{high} cells showed higher AXL expression in either transcript or protein levels. In the AXLi treatment study, the patients with *EGFR*^{WT} tumor in chemotherapy, untreated *EGFR*^{19Del} tumor, and resistant *EGFR*^{L858R} tumor are P29, P9 and P3, respectively. This information has been added into the caption of Fig. 6 in the revised manuscript.

Figure R8. AXL fluorescence intensity of 2-NBDG^{high} cell, C12R^{high} cells and double negative

from pleural effusion of P29. It is noted that all 2-NBDG^{high} cells (97) and C12R^{high} cells (44) were assayed and that shown in the figure. One hundred randomly selected double negative (2-NBDG^{low}/C12R^{low}/CD45^{neg}) cells were assayed and shown.

R2Q5. The study is based on the use of two different fluorescent probes to identify highly glycolytic vs mitochondrial oxidation cells. As the authors correctly point out in the discussion, it is possible that tumor cells growing in a tissue microenvironment may have different metabolic profiles compared with tumor cells growing in suspension in a pleural effusion. In fact, they show that the two metabolic phenotypes are different in cells growing in culture, where the two phenotypes correlate rather than being mostly mutual exclusive. Thus, it seems important for the authors to show that these same phenotype distributions exist in tumor cell populations isolated from tumor tissue – this would be relatively straightforward to do from either mouse NSCLC xenograft tumors/PDXs or cell isolated from lung tumor surgical resection samples.

Response: To investigate the single-cell metabolic phenotyping of tumor tissue, we collected lung adenocarcinoma surgical resection samples and digested them into a single cell suspension for subsequent 2-NBDG/C12R assay. Similar to the pleural effusion samples, 2-NBDG and C12R fluorescent levels of leukocytes present in the tumor tissues were used to generate the cut-offs of cells with high metabolic activity. Fig. R9 below shows all CD45^{neg} cells assayed on chip from tumor tissues of three lung adenocarcinoma patients. The numbers of cells with high and low metabolic activities are indicated in the figure. The results demonstrate the same mutual exclusive phenotype distributions that exist in cell populations isolated from tumor tissue.

Figure R9. Single-cell metabolic phenotyping of CD45 negative cells isolated from lung adenocarcinoma surgical resection tissues of three patients. Most of the metabolically active cells are differentially engaged in glycolysis or mitochondrial oxidation with very few double positive cells.

Change: In the revised manuscript, we added relevant discussion and included Fig. R9 in the SI

as Supplementary Fig 10.

R2Q6: The rationale for the use of the fluorescent probes is that they identify high glycolytic or mitochondrial oxidation cells. The Enrichr scores suggest that the subset of DEGS shared by the 5 patients profiled are enriched for genes associated with ETC and oxidative phosphorylation, however it is not clear if this actually marks the C12R cells that are supposed to use mitochondrial oxidation (the GSEA results don't confirm this). Since the use metabolic phenotyping forms the underlying premise of the entire study, it would be helpful if the authors can provide some explanation for this apparent contradiction. Also, it would be useful to know if there are other gene signatures that are enriched in the C12R cells besides epithelial genes (Fig 5D-F)?

Response and Change: Because one RNA-seq sample was mislabeled in the original data set, we re-performed all the analysis in Fig. 5 and updated the results.

The Enrichr results indeed showed enrichment signatures of ETC associated programs. The reason that we did not elaborate in detail these enriched metabolic programs in the original submission is that the associations between gene expression profiles and actual cellular metabolic functions are only remotely connected. The activities of both glycolysis and mitochondrial oxidation depend heavily on the post translational modifications of various metabolic enzymes, which cannot be fully revealed by the expression levels of metabolic genes. We therefore resorted to imaging analysis to confirm the co-localization of C12R signal and mitochondria plus a series of functional validations (metabolite perturbations, inhibitor perturbations, etc) to ensure C12R is indeed a readout of mitochondrial oxidation (Fig. 1c, e, f and Supplementary Fig. 2). In the revised manuscript, we further used the commercially available Seahorse assay to confirm that 2-NBDG and C12R readouts are consistent with extracellular acidification rate (ECAR) and oxygen consumption rate (OCR) readouts, respectively, when we perturb A549 cells with a series of metabolic inhibitors (2-DG, Oligomycin, Phenformin, BPTES, Etomoxir) to repress either glycolysis or mitochondrial respiration (See Figure R14 below, and Fig. 1f; Supplementary Fig. 2 in the revised manuscript). These results indicate that 2-NBDG and C12R are robust readouts for cellular glycolytic activities and mitochondrial oxidation, which fully supports the underlying premise of the entire study.

Per the reviewer's request, in the revised manuscript, we enriched the DEGs upregulated in each metabolic phenotype separately. The DEGs upregulated in C12R^{high} cells showed enrichment in ETC-related proton transporting (Fig. R10a). We further directly inspected the expression levels of relevant genes involved heavily in both glycolysis and mitochondrial oxidation. Consistently, 2-NBDG^{high} cells have higher expression in many glycolysis-related genes and lower expression in mitochondrial oxidation-related genes than C12R^{high} cells across different patients (Fig. R10b below and Fig. 5c,d in the revised manuscript). However, as we expected, there are some exceptions in certain genes for certain patients. For example, 2-NBDG^{high} cells from P10 have higher levels of glucose transporter and lactate dehydrogenase (*SLC2A3* and *LDH*) but lower levels of hexokinase (*HK1* and *HK2*) than C12R^{high} cells (Fig. R10b). Such mixed patterns are

presumably due to the fact that the changes in gene expression levels are only weakly and remotely connected with actual cellular metabolic activities. We therefore believe that, direct functional validation with comparison to gold standard metabolic assays (Seahorse measurements) is a more reliable way to confirm the metabolic functions indicated by the probes.

Other gene signatures enriched in the C12R cells have been listed in Supplementary Data 6.

Figure R10. a Enrichment of the DEGs up-regulated in each metabolic phenotype and shared by at least 4 out of 5 patients against 5 representative public databases by Enrichr. The top two entries ranked by the combined scores from each database are plotted. **b** Log₂ fold change of gene expression levels between the two metabolic phenotypes (2-NBDG^{high} vs C12R^{high}) across 5 patients. Representative genes involved in glycolysis, mitochondrial ETC complexes, and fatty acid oxidation are listed.

Minor points:

R2Q7. The figure legends for Figure 2 B and C are out of order.

Response and Change: We have corrected this mislabeling.

R2Q8. The authors show that AXL inhibition induces apoptosis in glycolytic but not mitochondrial oxidation cells. It would be helpful to have a more detailed description of how the on-chip apoptosis assay was actually performed since this seems different from the more standard flow cytometry assay that most readers will be familiar with. In Figure 6B-C, what exactly do the Y axis units mean (I assume that a.u. = absorbance units)? Is the cell viability determined by counting the number of apoptotic cells? What is the actual frequency of apoptotic cells with AXLi and vehicle? The authors should provide the raw values for how many cells were counted, how many were scored as apoptotic, for both AXLi and baseline. In most applications looking at cells treated with RTK inhibitors, 12 hours is typically too short to see apoptosis occurring after drug treatment, so these details are essential for demonstrating the robustness of the assay.

Response and Change: We have modified the Methods to present a more detailed description of AXL inhibition experiment and the apoptosis assay. Before AXL inhibitor treatment, we labeled metabolically active cells in pleural effusion after 5 min incubation with 2-NBDG and C12R probes. After on-chip washing with cold PBS and DMEM, the chip was sealed with a porous membrane to avoid cell loss in the following steps. All cells on the chip were imaged and then treated with the AXL inhibitor. At the end of inhibitor treatment, PE Annexin V Apoptosis Detection Kit was used to stain apoptotic cells on chip followed by the second round of imaging. For data analysis, we counted the metabolically active cells identified by 2-NBDG or C12R, and then calculated the percentage of cell apoptosis and viability in this cell population (formula 1 & 2). Relative viability was calculated based on formula 3. Per the reviewer's request, we provided the raw values of number of cells counted and scored in Fig. R11 below. We also included this data as a supplementary figure (Supplementary Fig. 15) in the revised manuscript. We did not use flow cytometry for apoptotic assay because we were targeting rare metabolically active cells rather than all of cells present in the pleural effusion. As we mentioned before, metabolically inactive cells included different cell types and unknown cell viability. In contrast, metabolically active cells were viable tumor cells in distinct metabolic phenotypes.

While 12 hour treatment is relatively short, tumor cells in pleural effusions are more fragile than cancer cell lines and therefore twelve hours of incubation caused apoptosis. Annexin V can detect early stage apoptosis events as well. Our data showed statistically significant differences in different treatment groups.

$$\% \text{ of apoptosis} = \frac{\text{apoptotic metabolically active cells}}{\text{total metabolically active cells}} \quad (1)$$

$$\text{Viability} = 1 - \% \text{ of apoptosis} \quad (2)$$

$$\text{Relative viability} = \frac{1 - \% \text{ of apoptosis (AXLi or control)}}{1 - \% \text{ of apoptosis (control)}} \quad (3)$$

a.u in Fig. 6b,c represents arbitrary unit

Figure R11. Metabolically active cells assayed and their viability under AXLi inhibition treatment in P3, P9 and P29. Each experiment had three replicates.

R2Q9. In the discussion, the authors state (lines 395-398) “We speculate that a very similar mechanism may contribute to ...” The authors may wish to reference a recent study that showed that EGFR cells surviving EGFR TKI exhibited a mesenchymal phenotype, could evolve genetic mechanisms of acquired resistance TKI, and then were more tolerant to subsequent treatment with 3rd generation EGFR TKI. (PMID: 26828195)

Response and Change: We thank the reviewer for this suggestion. We added that paper as a reference in the revised manuscript.

R2Q10. In the discussion (lines 415 – 419), the authors state “Therefore our results suggest that NSCLC patients with high N/R ratios should probably take more aggressive treatments...or potentially resort to checkpoint immunotherapy...” As these studies are preliminary and have not even been tested using in vivo models, the authors may wish to refrain from making statements that appear to be clinical recommendations. In particular, clinical studies have shown that immune checkpoint inhibitors are largely ineffective in EGFR mutant patients, even when PDL1

expression is high (PMID: 29874546).

Response: As the reviewer pointed out, in first-line therapy of advanced NSCLC, Lisberg et al. reported a lack of efficacy of Pembrolizumab in TKI naïve, PD-L1+, EGFR-mutant patients, including those with PD-L1 expression $\geq 50\%$ (Lisberg A et al., *J Thorac Oncol.* 2018, 13, 1138-1145). While the EGFR-mutant TKI naïve NSCLC patients, as a whole, may not benefit from first-line checkpoint inhibitor treatment, the subpopulation of *EGFR*-mutant patients who are innately resistant to the first-line EGFR TKI and with a high PD-L1 expression level might have potential benefit.

Our results indicate that newly diagnosed *EGFR*-mutant patients with predominant 2-NBDG^{high} tumor cells in their MPEs (high N/R ratios) are less likely to respond to the first-line EGFR-TKI. In the meantime, 2-NBDG^{high} cells are also having higher PD-L1 expression (Fig. 5f and Supplementary Fig. 13). Therefore, the metabolic phenotyping might be a predictive markers for identifying the *EGFR*-mutant patients with high PD-L1 expression who are innately resistant to EGFR TKI. This patient population might include potential beneficiaries to the anti-PD1 blockade. Of course, all these hypotheses need rigorous preclinical and clinical studies to validate.

Change: We re-wrote the relevant discussion to make our points clearer. Per the reviewer's suggestion, we tried to use careful and conservative languages in the discussion to avoid any misconception.

Review #3

R3Q1: The one suggestion would be to revise the title of the paper to reflect what was measured was a ratio between glycolysis and oxidative phosphorylation. As written, one gets the impression that it is an overall metabolic profiling.

Response and Change: We appreciate the reviewer's comment. Due to the strict length limitation of the title, it is challenging to make it very specific. However, in the revised abstract, we emphasize the point that we are focusing on the two metabolic phenotypes and the ratio between the two phenotypes is predictive for patient therapy response.

Review #4

R4Q1: Underrepresentation of Kras-driven lung adenocarcinoma. The percent of NBDG^{high} cells in the one KRAS mutant patient 4 is high, which would agree with previous literature. However, additional patients with KRAS mutations would provide more proof of a genotype-specific difference in the glucose vs mitochondrial oxidation phenotype of MPEs. This is particularly important because most patients are EGFR mutant and therefore the conclusion the authors make about the N/R score in lines 238-241 about the importance of the N/R score in predicting response of patients with different genotypes is not supported by the data.

Response and Change: The patient samples collected in this study are from China and only a small population of Chinese lung adenocarcinoma patients is bearing *KRAS* driver mutation. The reported frequencies of *KRAS* driver mutation in Chinese lung adenocarcinoma patients are around 6% - 8% (Gou LY and Wu YL, *Lung Cancer: Targets and Therapy*, 2014, 5, 1-9; Wang et al. *Oncotarget* 2015, 6, 34300), which is a lot lower than the 26% frequency in Europe (Boch C. et al., *BMJ Open*, 2013, 4, e002560). In our study, we did not rule out MPE samples of patients with *KRAS* mutation during our sample collection, but failed to obtain a large number of MPE samples of *KRAS* mutation-driven patients. During the initial submission and manuscript revision, we received another two *KRAS*-mutant patients samples and we added them in the revised manuscript. One patient has follow-up evaluation and has been included in the survival analysis. The other patient sample was very recent, therefore only chip-based MPE analysis was performed with calculated N/R ratio without follow-up evaluation data available. Since the manuscript revision cannot be longer than 6 months, we are unable to add more *KRAS*-mutant patients samples in our cohort.

As shown in Fig. 2a, the distribution of the mutational profiles in our patient cohort is consistent with the frequencies of various mutations observed in the clinic. Among the three *KRAS*-mutant patient samples, two of them (P4, P30) are showing glycolytic phenotype with high percentages of 2-NBDG^{high} cells. One patient (P32) is having more C12R^{high} cells. Given the small sample size, it is difficult to conclude whether a genotype-specific difference in the glycolysis vs mitochondrial oxidation phenotype exists in *KRAS*-mutant patients. Additionally, although we did not see any exception in our dataset, the predictive capability of the N/R ratio for patients with low frequent mutation types still needs larger patient cohort to validate. We revised the discussion about “the importance of the N/R score in predicting response of patients with different genotypes” to caveat this statement.

R4Q2: The authors should describe and graph the percentage of cells of each 4 populations in every patient. Based on the numbers for the CD45neg cells in patient 1, it appears that very few cells of the total (~500k cells) are actually assayed. Based on Fig2D, this seems to be the case for most patients.

Response: All nucleated cells on the chip (~500K cells) were assayed with 2-NBDG, C12R and CD45 for each patient. These cells were classified into two categories. One category was CD45+ cells, namely leukocytes, that usually constituted a majority of the cell population on the chip. The 2-NBDG and C12R levels of CD45+ cells were used to calculate the cut-offs for identifying metabolically active cells (red histograms in Fig. R12 below (i.e. Fig. 2b)). The other category was CD45- cells, including tumor cells, mesothelial cells, circulating epithelial cells, and other cell types. Metabolically active cells were identified in this category based on the cut-offs of 2-NBDG and C12R determined through leukocytes.

For example, in the case of P1, there were a total of 66 metabolically active cells, ~7,000 2-NBDG^{low}/C12R^{low}/CD45- cells, and ~490,000 CD45+ leukocytes. It is easy to precisely count the cell numbers for 2-NBDG^{high}/C12R^{low} or 2-NBDG^{low}/C12R^{high} or 2-NBDG^{high}/C12R^{high} cells, because their high fluorescence intensities facilitate the cell recognition and signal quantification.

However, it is technically challenging to measure and count 2-NBDG^{low}/C12R^{low}/CD45⁻ cells due to their low fluorescence intensities. The 2-NBDG^{low}/C12R^{low}/CD45⁻ cell population was not used in any subsequent analysis and was irrelevant to any major conclusions of this paper, we therefore skip documenting the precise cell numbers of this population for every patient. The cell numbers of all other three populations (out of the 500K input cells) for each patient are precisely documented in Table 1.

Figure R12. Scatter plot reports 2-NBDG and C12R fluorescence intensity of all CD45⁻ cells in MPE sample from P1. The histograms of 2-NBDG and C12R intensities of CD45⁺ leukocytes (red) in MPE are shown on the top and right to generate cut-offs for identification of 2-NBDG^{high} and C12R^{high} cells (black dots). 2-NBDG^{high} and C12R^{high} cells are gated out by five and three standard deviations above mean of CD45⁺ leukocytes, respectively. 2-NBDG^{low}/C12R^{low}/CD45⁻ cells are displayed in blue dots.

R4Q3: What is the explanation for some CD45^{neg} cells from patient 1 not having the EGFR ex19 deletion?

Response and Change: This question is in line with R2Q3. The cells with wild type driver oncogene may be derived from intratumor heterogeneity or be attributed to the limitations of single-cell genome amplification technique. But they are indeed tumor cells that can be verified

by genome-wide copy number variation (CNV) analysis.

For example, in Fig. R7 below, 16 metabolically active cells from P1 were isolated, followed by single-cell genome amplification and Sanger sequencing. Thirteen of them (~81%) were found to harbor *EGFR*^{19Del} and the remaining three cells exhibited wild type *EGFR* (one 2-NBDG^{high} and two C12R^{high} cells). To determine the malignancy of the three *EGFR*^{WT} cells, we performed whole-genome sequencing (~0.1× sequencing depth) of all sixteen metabolically active cells and three white blood cells to characterize single-cell copy number (CNV) patterns (Fig. R7). CNV allows us to survey the entire genomic landscape and determine the malignancy based on chromosomal gains and losses. Two *EGFR*^{WT} cells exhibited gain and loss CNV patterns consistent with other *EGFR*^{19Del} cells, which can be identified as tumor cells. While one *EGFR*^{WT} cell showed distinct global CNV profile from other cells, its dramatic copy number alteration across many chromosomal regions (compared to WBCs) suggested a high likelihood of malignancy. It is presumably derived from a different colony of the primary tumor due to the intratumoral heterogeneity.

Additionally, in our previous PNAS paper (Tang et al., *PNAS*, 2017, 114, 2544-2549), we performed the whole exome sequencing (WES) on *KRAS*^{WT} cells isolated from the MPE sample of a *KRAS* mutant patient. We screened the mutations with the Qiagen's Lung Cancer Panel, containing 45 most relevant driver oncogenes and tumor suppressor genes in lung cancer. These *KRAS*^{WT} cells contained alterations in several marker oncogenes and tumor suppressors, including *BRAF*, *EGFR*, *PIK3CA*, *PTEN*, and *TP53*. These data also suggest the malignancy involvement of the *KRAS*^{WT} cells.

In addition to the data shown above, we performed CNV analyses of patient MPE samples in many pilot studies during the development of the on-chip metabolic cytometry platform. The accumulated results suggest that most of the WT/CD45^{neg} metabolically active cells are indeed tumor cells with CNV patterns consistent to the cells bearing driver oncogene mutations.

Figure R7. The single-cell CNV profiles and detected *EGFR* mutation status of metabolically active cells and white blood cells from P1.

R4Q4. Why is there a discrepancy between MPE and cancer cell line data? The authors should elaborate. Is this simply the *in vivo* vs *in vitro* growth conditions? When the authors culture cells for viability assays in Figure 6B-C, do they see any change in the N/R ratio?

Response and Change: Discrepancy between cancer cell lines and clinical biospecimens are commonly seen and have been widely reported. Such discrepancy may be attributed to the following reasons. (1) Cancer cell lines (A549, H1975, etc.) used in this paper are well established and have been passaged for many times. They are relatively homogeneous cell population that is capable of self-replication in standard cell culture medium. But cancer cells present in patient MPE samples are highly functionally heterogeneous in metabolic activity, metastatic propensity and immune response. Such heterogeneity may derived from differences in genetic and epigenetic states as well as in growth history. (2) Cancer cell lines are cultured *in vitro* in homogeneous cell culture medium. The growth environment is the same for all the cells. However, the cancer cells present in MPE samples may be derived from different parts of the primary tumor at different time points. Therefore, the microenvironment differences will also contribute to the observed discrepancy between MPE and cancer cell lines. We modified the relevant text to elaborate more discussion on this discrepancy per reviewer's suggestion.

For the viability assays shown in Fig. 6b,c, we used fresh patient tumor cells present in the MPE samples without any *in vitro* culture/expansion. In other words, we used the same MPE samples for both the metabolic phenotyping and the AXLi viability assay. The viability assay was performed in the microwell chip at the single-cell level. So, we do not need to grow the cells to enough number for the viability assay. Therefore, the N/R ratio was the same for the cell population used for viability assay. We did not re-measure the N/R ratio after AXLi treatment. However, since AXLi induced significant apoptosis in 2-NBDG^{high} subset rather than C12R^{high} subset, the N/R ratio is supposed to be decreased. We have modified the Methods to present a more detailed description of the viability assay. Please also see the response to R2Q8.

R4Q5. How does the AUC of the ROC curve compare to other known markers of MPEs previously identified?

Response: The ROC curve from the PLS-DA in this study is for predicting patient therapy responses. To our best knowledge, no previous work has attempted to integrate various markers of MPEs to predict patient therapy responses. Most previous MPE markers are set for distinguishing MPEs from benign pleural effusion, such as Light criteria (Light RW, *N. Engl. J. Med.* 2002, 346, 1971-1977). But attempts have been made to predict survival based on the clinical characteristics of pleural fluid. In a recent study (Clive AO, *et al.* Predicting survival in malignant pleural effusion: development and validation of the LENT prognostic score, *Thorax* 2014, 69, 1098–1104), performance status (ECOG score), lactate dehydrogenase (LDH) level in the pleural fluid, neutrophil to lymphocyte ratio, and tumor type were combined to determine a LENT score for predicting patient survival (See the table below). This LENT score was found to

be better than performance status (ECOG) alone for survival prediction. The most important factor for this score calculation is the tumor type, which is a fixed parameter in our study (i.e. stage IV LADC). Based on the LENT score, most of our patients are falling into the moderate risk population. However, the metabolic phenotyping assay of MPE, particularly the N/R ratio, can further predict the responders and nonresponders as well as their survival in this moderate risk patient population identified by LENT score. Therefore, the metabolic phenotyping assay can make much finer patient segregation than LENT score.

Table 3 The LENT score calculation

	Variable	Score
L	LDH level in pleural fluid (IU/L)	
	<1500	0
	>1500	1
E	ECOG PS	
	0	0
	1	1
	2	2
	3-4	3
N	NLR	
	<9	0
	>9	1
T	Tumour type	
	Lowest risk tumour types	0
	▶ Mesothelioma	
	▶ Haematological malignancy	
	Moderate risk tumour types	1
	▶ Breast cancer	
	▶ Gynaecological cancer	
▶ Renal cell carcinoma		
Highest risk tumour types	2	
	▶ Lung cancer	
	▶ Other tumour types	
Risk categories	Total score	
Low risk	0-1	
Moderate risk	2-4	
High risk	5-7	

ECOG PS, Eastern Cooperative Oncology Group performance score; LDH, lactate dehydrogenase; NLR, neutrophil-to-lymphocyte ratio.

R4Q6. The authors present data that suggests that driver mutations, in particular EGFR mutations or CNVs (in 3 patients) are insufficient to explain the observed uncoupling of metabolic phenotypes in patients MPE samples. However instead of using those same 3 patients to see if gene expression can explain the metabolic phenotypes in the context of no oncogenic driver and CNV, they go on to analyze a random set of 5 patient MPEs by RNAseq. Are the gene

expression differences in those 5 random MPE samples due to differences in oncogenic driver (eg Kras, EGFR) or CNVs? Furthermore, additional mutations, independently of the known major oncogenic driver may be leading to metabolic rewiring of tumor cells. The authors should use their RNAseq data to call protein-altering mutations in the MPE samples. Furthermore, they should elaborate on the OXPHOS genes that are enriched in Fig5C. Given that they are able to stain cells from MPEs by immunofluorescence, they should check the levels of recurrently altered OXPHOS genes by IF.

Response and Change: There are practical reasons that we are unable to analyze the gene expression profile of cells from the same 3 patients for the CNV analysis. The number of metabolically active cells in each patient sample is very limited (sometimes less than 10 for a specific metabolic phenotype). Simultaneous extraction of DNA and RNA from the same single cells is technically challenging. So, we have to use different cells for DNA and RNA extraction, respectively. The successful rates of the whole genome amplification of single cell genome as well as the SMART-seq amplification of cDNAs from extremely small number of cells are also below 100% for clinical samples. Therefore, the number of metabolically active cells from a single patient is often not sufficient for both CNVs and RNA-seq analyses.

The five patients for RNA-seq analysis include three *EGFR*^{19Del} patients and two wild type patients. The gene expression differences across patients may certainly be influenced by the differences in their genetic background. This is also the reason for us to focus on the DEGs between the two metabolic phenotypes that are shared across at least 4 out of 5 patient samples. These shared DEGs are less likely to be induced by the differences in patient genetic background and may represent common molecular signatures beyond the genetic level. Per the reviewer's request, in the revised manuscript, we further elaborated the expression levels of relevant genes involved heavily in both glycolysis and mitochondrial oxidation. We found relatively consistent up-regulation of glycolysis-related genes in 2-NBDG^{high} cells and up-regulation of ETC complexes-related genes in C12R^{high} cells across different patients (see Fig. R10b in R2Q6 or Fig. 5c in the revised manuscript). However, as we expected, there are some exceptions in certain genes for certain patients. For example, 2-NBDG^{high} cells from P10 have higher levels of glucose transporter and lactate dehydrogenase (*SLC2A3* and *LDH*) but lower levels of hexokinase (*HK1* and *HK2*) than C12R^{high} cells (Fig. R10b in R2Q6 or Fig. 5c in the revised manuscript). Such mixed patterns are presumably due to the fact that the changes in gene expression levels are only weakly and remotely connected with actual cellular metabolic activities. We did not identify any preference to certain driver oncogene in the gene expression levels of these metabolic-related genes.

Per the reviewer's suggestion, we called the protein-altering mutations using the RNA-seq data for the five patients and included the relevant analysis in the revised manuscript. However, we did not identify any protein-altering mutation incurred in a specific metabolic phenotype that was shared across all five patients (Supplementary Data 2). We further inspected the nonsynonymous (possibly damaging) mutations in a specific metabolic phenotype that were shared by four out of the five patients. These mutations have not been reported to be related to any known metabolic

regulations pursuant to the two metabolic phenotypes (Supplementary Data 2). Therefore, it is less likely that the observed metabolic phenotypes are derived from a mutation-induced metabolic rewiring. We observed that the two metabolic phenotypes are linked to the EMT program where 2-NBDG^{high} cells are expressing more mesenchymal markers. Activation of glycolysis by EMT has been observed in many cancer types, where it is required for both cytoskeleton remodeling and increasing cell traction (Shiraishi T, et al., *Oncotarget*, 2015, 6, 130-143; Masin M, et al, *Cancer Metab.* 2014, 2:11). Therefore, we speculate that the epigenetic reprogramming associated with EMT might contribute to the observed metabolic program reliance.

We did not perform IF staining to check the levels of mitochondrial oxidation related genes for two reasons. (1) The activities of both glycolysis and mitochondrial oxidation depend heavily on the post translational modifications of various metabolic enzymes, which cannot be fully revealed by the protein expression levels of a few metabolic genes. This is also the reason that we performed imaging analysis to confirm the co-localization of C12R signal and mitochondria plus a series of functional validations (metabolite perturbations, inhibitor perturbations, etc) to ensure C12R is indeed a readout of mitochondrial oxidation (Fig. 1c, e, f and Supplementary Fig. 2). (2) To our best knowledge, no single gene or a small set of genes can robustly resolve the metabolic phenotypes than the metabolic functional staining itself. In the revised manuscript, we further used the commercially available Seahorse assay to confirm that 2-NBDG and C12R readouts are consistent with extracellular acidification rate (ECAR) and oxygen consumption rate (OCR) readouts, respectively, when we perturb A549 cells with a series of metabolic inhibitors (2-DG, Oligomycin, Phenformin, BPTES, Etomoxir) to repress either glycolysis or mitochondrial respiration (See Figure R14 below, and Fig. 1f; Supplementary Fig. 2 in the revised manuscript). These results indicate that 2-NBDG and C12R are robust readouts for cellular glycolytic activities and mitochondrial oxidation.

R4Q7: The increased levels of AXL in the N/R high cells is interesting, particularly because of data in Fig6B-C showing a response of these cells to AXL inhibition. However, what remains unclear is whether AXL has a functional role in metabolic rewiring of cells, leading to an increase of the glycolysis over oxphos. Using the experimental setup in figure6 the authors should assess the following:

- 1) Does AXL inhibition lead to a decrease in the N/R ratio in 2-NBDG^{high}/C12R^{low} cells as compared to 2-NBDG^{low}/C12R^{high} cells?
- 2) the prediction would be that 2-NBDG^{high}/C12R^{low} cells would be more sensitive than 2-NBDG^{low}/C12R^{high} cells to metformin or other mitochondrial inhibitors.
- 3) Are 2-NBDG^{high}/C12R^{low} cells more sensitive to 2DG or other glycolytic inhibitors as compared to 2-NBDG^{low}/C12R^{high} cells.
- 4) Are 2-NBDG^{low}/C12R^{high} cells more reliant on alternative sources of carbon other than glucose to drive TCA cycle and OXPHOS? They should test whether 2-NBDG^{low}/C12R^{high} cells are more sensitive to inhibition of glutamine catabolism by glutaminase inhibition (eg BPTES,

CB839).

Response and Change: While a detailed mechanistic study of the functional role of AXL in the metabolic rewiring is beyond the scope of this manuscript, the strong association between AXL up-regulation and elevated glycolysis have been reported in other cancer types. For example, Sadahiro et al., showed that downregulation of PDGF α drives AXL expression accompanied by increased glycolysis in mesenchymal GBM subtype (Sadahiro et al., *Cancer Research*, 2018, doi: 10.1158/0008-5472.CAN-17-2433). Cheng et al., also found that the AXL/TNS2/IRS-1 cross-talk can up-regulated GLUT4/PDK and play a critical role in glucose metabolism in pancreatic cancer. (Cheng et al., AXL phosphorylates and up-regulates TNS2 and its implications in IRS-1-associated metabolism in cancer cells, *Journal of Biomedical Science*, 2018, 25:80). Therefore, we speculate that AXL may play a similar functional role here in prompting the glycolytic phenotype.

1) We did not re-measure the N/R ratio after AXLi treatment. However, since AXLi induced significant apoptosis in 2-NBDG^{high} subset compared to C12R^{high} subset, the N/R ratio is supposed to be decreased.

2-4) To address the reviewer's questions, a series of inhibitors related to metabolism were used to test the susceptibility of two metabolic subsets, including glycolytic inhibitor (2-DG), mitochondrial oxidation inhibitors (phenformin), glutaminase inhibitors (BPETS) and fatty acid oxidation inhibitor (etomoxir). Specifically, pleural effusion of Patient 30 was filtered, followed by red blood cells removal and incubation with APC-conjugated anti-CD45 for 30 min. After washing and cell counting, we used 2-NBDG/C12R to fluorescently flag metabolically active cells in ~500,000 cells via a 5-min rapid incubation. After on-chip washing with cold PBS and DMEM, the chip was sealed with a porous membrane to avoid cell loss in the following steps. All cells on the chip were imaged and then treated with the a series of inhibitors (2-DG: 5 mM; phenformin: 25 μ M; BPETS: 10 μ M; etomoxir: 200 μ M) for 12 h at 37 \square with DMSO as the control. At the end of inhibitor treatment, PE Annexin V Apoptosis Detection Kit I was used to stain apoptotic cells on chip followed by the second round of imaging. For data analysis, we counted the metabolically active cells identified by 2-NBDG and C12R, and then calculated the percentage of cell apoptosis in this cell population (formula 1). Relative viability was calculated based on formula 2.

$$\% \text{ of apoptosis} = \frac{\text{apoptotic metabolically active cells}}{\text{total metabolically active cells}} \quad (1)$$

$$\text{Relative viability} = \frac{1 - \% \text{ of apoptosis (inhibitor)}}{1 - \% \text{ of apoptosis (control)}} \quad (2)$$

Table R2. Numbers of two metabolically active cell subpopulations in MPE from Patient 30 in the inhibition experiments. For each inhibitor, two replicates were performed.

	# of 2-NBDG ^{high} cell assayed	# of C12R ^{high} cell assayed
--	---	---

2-DG	126	31
	152	34
Phenformin	125	31
	133	30
BPETS	137	26
	158	32
Etomoxir	129	27
	116	23
DMSO	144	30
	122	26

As shown in the Fig. R13 below, 2-NBDG^{high} cells are more sensitive to the inhibition of glycolysis by 2-DG as compared to C12R^{high} cells. On the contrary, C12R^{high} cells are more sensitive to inhibition of mitochondrial oxidation by phenformin and inhibition of fatty acid oxidation by etomoxir as compared to 2-NBDG^{high} cells. As for BPETS, no significant difference is observed between two subsets. The results are consistent with the observed up-regulation of genes involved in glycolysis in 2-NBDG^{high} cells and up-regulation of genes involved in mitochondrial oxidation and fatty acid beta-oxidation in C12R^{high} cells (Fig. 5c). We included these results in the revised manuscript.

Figure R13. Relative viability of two metabolically active cell subpopulations in response to

2-DG (5 mM), phenformin (25 μ M), BPETS (10 μ M), and etomoxir (200 μ M) with respect to the DMSO control (* $p < 0.05$; ** $p < 0.005$). Two replicates were performed in each experiment.

R4Q8: Given that the authors are able to grow enough cells from MPEs to perform viability assays, it is critical that the authors use alternative methods to validate their N/R scores using a different platform. The use of seahorse would enable them to use low cell numbers and measure oxygen consumption and glycolysis (media acidification).

Response and Change: There is a misconception about how the viability assays were performed. We apologize that we did not make it clear in the original manuscript. We used fresh patient tumor cells present in the MPE samples for the viability assays without any *in vitro* culture/expansion. This is because that culturing primary cells *in vitro* may alter their phenotypic and functional behaviors. The viability assays were performed in the microwell chip at the single-cell level. So, we do not need to grow the cells to enough number for the viability assays. We have modified the Methods to present a more detailed description of AXL inhibition experiment and the viability assay (Please also see R2Q8).

Seahorse is a bulk-level assay that cannot be used to assay the rare tumor cells (<100 cells) at the single-cell level. For this reason, Seahorse cannot be directly used for validating N/R scores as an alternative method. However, Seahorse assays can indeed be utilized to further validate 2-NBDG and C12R as the readouts for glycolysis and mitochondrial oxidation, in addition to the investigations presented in original Fig. 1. As suggested by the reviewer, we used a series of inhibitors to perturb the glycolysis and mitochondrial oxidation of a lung cancer cell line (A549 cells), including glycolytic inhibitor (2-DG), mitochondrial oxidation inhibitors (oligomycin, phenformin), glutaminase inhibitors (BPETS) and fatty acid oxidation inhibitor (etomoxir). Specifically, 10,000 A549 cells were plated in the seahorse cell plate and incubated overnight with RPMI 1640 medium supplemented with 10% FBS. A549 cells were then treated with inhibitors (2-DG: 5 mM; oligomycin: 2 μ M; phenformin: 25 μ M; BPETS: 10 μ M; etomoxir: 200 μ M) with DMSO as a control in fresh complete cell culture media for 12 hours. Thirty minutes before the assay, the media was changed to fresh media containing inhibitors at the same concentrations. Extracellular Acidification Rate (ECAR) and Oxygen Consumption Rate (OCR) were measured on a XFe96 Seahorse Biosciences Extracellular Flux Analyzer for 10 cycles.

The results confirm that the changes in 2-NBDG and C12R measured by our on-chip metabolic cytometry are fully consistent with those in ECAR and OCR readouts measured by the Seahorse assay, respectively, across various metabolic perturbations (see Fig. 14R below). These results are now being included in Fig. 1f and Supplementary Fig. 2d of the revised manuscript.

Figure R14. a,b Changes of ECAR and OCR of A549 cells in response to 2-DG, oligomycin, phenformin, BPETS, and etomoxir relative to the DMSO control. Three replicates were performed in each experiment and each replicate represented the average of ten cycles of ECAR and OCR measurements. Data are presented as the mean \pm SD. **c** Relative 2-NBDG, ECAR, C12R, and OCR readout changes of A549 cells in response to a set of metabolic inhibitors with respect to DMSO control. Data are presented as the mean \pm SD. For ECAR and OCR measurements, three replicates were performed in each experiment and each replicate represented the average of ten cycles of ECAR and OCR measurements. For 2-NBDG and C12R measurements, more than 5,000 cells were assayed at the single-cell level in each condition.

Minor comment:

R4Q9: Line 236 the authors should correct viable to variable.

Response and Change: We have corrected this typo.

Reviewers' Comments:

Reviewer #1:

Remarks to the Author:

The authors have answers my queries, and I have no further comments.

Reviewer #2:

Remarks to the Author:

In this resubmission, the authors present a revised manuscript that addresses all of my comments from the initial submission. In particular, they should be commended for adding additional patients and new analysis that supports their findings that an increased proportion of glycolytic mesenchymal cells in malignant pleural effusions portends a worse treatment response and prognosis. My only remaining minor comment is that a bit more clarification about the on-treatment collection would be helpful – particular about the EGFR mutant patients (who comprise the largest subset of patients and may have the most differential response to TKI vs chemotherapy). Particularly, it would be helpful to the reader to indicate (perhaps in a supplemental version of Figure 4b) whether the on-therapy samples were taken early in treatment when patients were responding, or late in the course of therapy when the disease was starting to progress. My concern is not about the conclusions, but rather to help the reader consider the biology that might underly these findings in relation to treatment response and progression.

Reviewer #3:

Remarks to the Author:

I have no further comments

Reviewer #4:

Remarks to the Author:

Authors have adequately addressed most of this reviewers major concerns.

We deeply appreciate all the reviewers for their insightful and constructive comments that significantly improved our manuscript. Please find below the point-by-point responses to reviewers' questions.

Reviewer #1 (Remarks to the Author):

The authors have answers my queries, and I have no further comments.

Reviewer #2 (Remarks to the Author):

In this resubmission, the authors present a revised manuscript that addresses all of my comments from the initial submission. In particular, they should be commended for adding additional patients and new analysis that supports their findings that an increased proportion of glycolytic mesenchymal cells in malignant pleural effusions portends a worse treatment response and prognosis. My only remaining minor comment is that a bit more clarification about the on-treatment collection would be helpful – particular about the EGFR mutant patients (who comprise the largest subset of patients and may have the most differential response to TKI vs chemotherapy). Particularly, it would be helpful to the reader to indicate (perhaps in a supplemental version of Figure 4b) whether the on-therapy samples were taken early in treatment when patients were responding, or late in the course of therapy when the disease was starting to progress. My concern is not about the conclusions, but rather to help the reader consider the biology that might underly these findings in relation to treatment response and progression.

Response: We appreciate this constructive comment. In the revised manuscript, we included a new column in the Supplementary Table 1 to outline the responses to the ongoing therapies for on-therapy patients at the time of the MPE collection and metabolic phenotyping. In addition, per the reviewer's suggestion, we included a new supplementary figure (Supplementary Fig. 11) to illustrate the therapy responses at the time of MPE draw for on-therapy patients.

Reviewer #3 (Remarks to the Author):

I have no further comments

Reviewer #4 (Remarks to the Author):

Authors have adequately addressed most of this reviewers major concerns.